# Learning to Defer
# on Anonymously-Annotated Data

## Abstract

Recent advancements in machine learning have prompted the development in human-machine cooperation to leverage the efficiency of machines and the reliability of human expertise. One such approach is *learning to defer* (L2D), where a model learns to selectively defer decision-making to humans based on their historical performance on labelled data. Traditional L2D methods require the same set of human experts in both training and deployment phase, so that the system can leverage their historical performance to allocate queries accordingly. This human-specific nature, however, renders inflexibility in dynamic real-world environments where expert availability can fluctuate due to leave, retirement, or the integration of new team members. To address this challenge, we propose leveraging anonymously-annotated datasets, which are commonly available in practice, to infer annotation patterns and cluster human annotators based on behavioural similarities. Building upon the clustering of human experts, we develop a variant L2D, known as L2D-Clusters, that defers queries to a cluster rather than a specific expert, with one expert from the cluster randomly selected to make the final decision. Empirical results show that our clustering aligns with known annotator behaviour and that L2D-Clusters performs comparably to expert-specific L2D, especially in onboarding scenarios with limited annotator-identified data.

## 1 Introduction

The rapid development of machine learning has led to several solutions to automate complex workflows to improve efficiency. While these solutions demonstrate exceptional capacity for processing a vast amount of information, they can also exhibit limitations, including susceptibility to biases and diminished reliability when dealing with uncertainty. *Learning to defer* (L2D) (Madras et al., 2018) – an extension of *learning to reject* (Chow, 1957) – has been introduced to address this weakness by explicitly integrating human experts into the decision-making pipeline. In general, an L2D system analyses a sample and decides to produce an automated decision or defer to one of the available human experts. This synergistic approach capitalises on the computational efficiency of machine learning models while leveraging the reliability of human experts, resulting in an optimal workflow with a minimal operational cost.

Learning to Defer systems are designed to analyse the training data annotated by human experts to identify the annotation pattern of each expert. This enables L2D to strategically assign samples to the experts that are more likely to achieve high predictive accuracy while simultaneously managing operational cost. Crucially, the efficacy of L2D is predicated on the availability of meticulously curated training datasets where **each annotation is explicitly associated with the human expert who provided it** (i.e., the *annotator-identified data*) (Madras et al., 2018; Mozannar & Sontag, 2020; Verma & Nalisnick, 2022; Nguyen et al., 2025). Demanding such detailed tracking of experts and annotations may, however, be impractical because of privacy concerns. In addition, many publicly available datasets often include annotations aggregated from multiple annotators, typically in the form of summary statistics such as label distributions (e.g., multinomial vectors), with no record of individual annotator identities, such as Treeversity (https://zooniverse.org/projects/friedmaw/treeversity), or Galaxy Zoo 2 (Willett et al., 2013). While these anonymously-annotated datasets are abundant and readily accessible, they are incompatible with standard L2D methods, which rely on explicitly attributed expert labels to learn individual deferral strategies. As a result, existing L2D approaches overlook this rich data source entirely. This limitation motivates the development of new methods

capable of leveraging anonymously-annotated data to reduce the annotation burden in training L2D systems, while retaining the benefits of expert-informed decision-making.

Another notable limitation of existing L2D methods stems from their nature of human-specific deferral (Mozannar & Sontag, 2020; Verma & Nalisnick, 2022; Tailor et al., 2024; Mao et al., 2023; Nguyen et al., 2025), meaning that *exactly the same* set of human experts available during training must also be present at deployment time for the system to function correctly. Such a requirement poses a significant practical challenge because of real-world fluctuations in personnel due to absence, retirement, or the onboarding of new annotators. Additionally, such a precise individual deferral is debatable, considering that human experts highly agree when annotating samples (Nowak & Rüger, 2010). This suggests that the critical factor in L2D lies at the shared annotation characteristics or competencies of human experts, not their unique identities. It, therefore, motivates developing L2D further by transitioning from a human-specific deferral mechanism towards a cluster-based approach, where deferral targets groups of experts who share comparable annotation profiles.

We, therefore, propose a probabilistic framework to address the outlined challenges. Specifically:

- We leverage anonymously annotated data to learn a model capable of clustering experts based on their annotation behaviours.
- We extend existing L2D methods from a human-specific paradigm to a cluster-based approach, wherein deferral decisions are made toward groups of experts rather than individuals.
- We also introduce a novel benchmark that simulates the expert onboarding process, wherein the training set comprises a limited number of expert annotations for evaluation purposes.

Empirical results demonstrate that the proposed expert clustering method exhibits strong alignment with known behavioural distinctions among annotators across both synthetic and real-world datasets. Furthermore, evaluations on the newly introduced benchmark reveal that our framework, which includes expert grouping and cluster-based L2D methods, consistently outperforms existing human-specific and human-agnostic L2D methods.

## 2 PROBLEM STATEMENT

Let an anonymously-annotated dataset without ground truth labels be denoted as $\mathcal{S} = \{(\mathbf{x}_n, \mathbf{t}_n)\}_{n=1}^{N}$, where $\mathbf{x}_n \in \mathcal{X} \subseteq \mathbb{R}^D$ denotes an input sample, and $\mathbf{t}_n \in \mathcal{T}_{L_n} = \{\mathbf{t}_n : \mathbf{t}_n \in \{0, 1, \ldots, L_n\}^C \wedge \mathbf{1}^\top \mathbf{t}_n = L_n\}$ is a multinomial variable representing the count of annotations made by $L_n$ human experts over $C$ categories. For example, $\mathbf{t}_n = \begin{bmatrix} 3 & 5 & 0 \end{bmatrix}^\top$ denotes the annotations of a 3-way classification (i.e., $C = 3$) made by 8 annotators (i.e., $L_n = 8$), in which three annotators label $\mathbf{x}_n$ as class 1, the other five annotators label as class 2, and none labels as class 3.

In addition, there are $M$ small *identified* sets made by $M$ human experts, each is denoted as $\mathcal{I}_m = \{(\mathbf{x}_j^{(m)}, t_j^{(m)})\}_{j=1}^{N_m}, m \in \{1, \ldots, M\}$, where $t_j^{(m)} \in \{1, \ldots, C\}$ is the annotation of the instance $\mathbf{x}_j^{(m)}$ labelled by the human expert indexed by $m$, and $N_m \ll N, \forall m \in \{1, \ldots, M\}$. In our setting, we only need an *identified* set $\mathcal{I}_m$ containing annotations made by the human expert indexed by $m$. This *identified* set is different from *context* sets used in other studies (Tailor et al., 2024; Strong et al., 2025), which demand the ground truth associated with each sample.

Our aim is to employ the anonymously-annotated dataset $\mathcal{S}$ to learn a model that can cluster annotation patterns into $K$ groups. Such a model can then be used to process each of the small data subset $\mathcal{I}_m$ to assign the cluster membership of the human expert indexed by $m$.

## 3 METHODOLOGY

### 3.1 MODEL ANONYMOUSLY ANNOTATED DATA

We employ the *inference of population structure* (Pritchard et al., 2000) in genetic and genomic research or the *latent Dirichlet allocation* (Blei & Jordan, 2003) in topic modelling to model the data generation process of each sample in the anonymous annotation dataset $\mathcal{S}$. Specifically, one sample $\mathbf{t}_n | \mathbf{x}_n$ in our setting is an analogy to one document in topic modelling. Under this analogy, each anonymous annotation $\mathbf{t} | \mathbf{x}$ is generated from one of $K$ "topics" or clusters of human experts. The process to generate anonymous annotations can be described as follows:

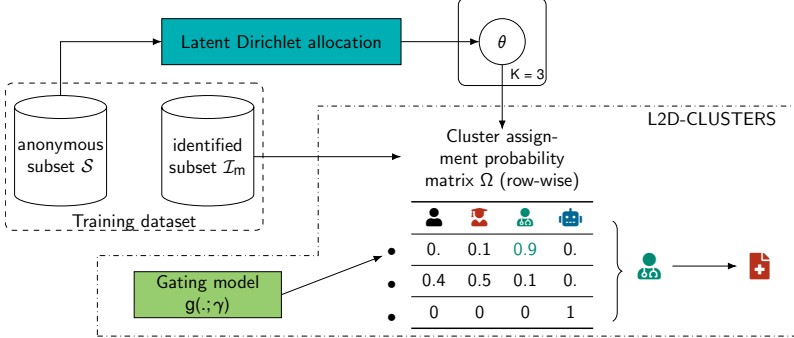

Figure 1: Diagram illustrating the proposed two-stage clustering-based L2D, where the first stage trains a clustering model on anonymous annotation data (top left part), and the second stage performs learning-to-defer to clusters of human experts (bottom right part in dash-dotted frame).

1. draw a sample from data distribution: $\mathbf{x} \sim \Pr(\mathbf{x})$,

2. draw the probability vector of clusters being selected from a Dirichlet prior: $\mathbf{u} \sim \mathrm{Dir}(\mathbf{u}|\alpha)$,

3. each annotation of the sample $\mathbf{x}$ is generated as follows:

   (a) draw a categorical variable to select a cluster: $\mathbf{z} \sim \mathrm{Categorical}(\mathbf{z}|\mathbf{u})$,

   (b) draw an annotation from that cluster: $\mathbf{t} \sim \Pr\big(\mathbf{t}|\mathbf{x}, \mathbf{z}, (\theta_k)_{k=1}^K\big) = \mathrm{Categorical}(\mathbf{t}|h(\mathbf{x}; \theta_\mathbf{z}))$,

where $\alpha \in \mathbb{R}_+^K$ is the concentration parameter of the Dirichlet distribution, $h(.; \theta_k) : \mathcal{X} \to \Delta_{C-1}$ is a model, parameterised by $\theta_k$, representing human experts belonging to a group indexed by $k$, with $\Delta_{C-1} = \{\mathbf{v} : \mathbf{v} \in [0,1]^C \wedge \mathbf{1}^\top \mathbf{v} = 1\}$ denoting the $(C-1)$-dimensional probability simplex. The data modelling can also be visualised as the graphical model shown in Fig. 4 (see Appendix C).

The objective of this modelling approach is to learn the set of parameters $(\theta_k)_{k=1}^K$ of the clustering model by maximising the likelihood on the observed anonymous annotation dataset as follows:

$$\max_{(\theta_k)_{k=1}^K} \sum_{n=1}^N \ln \Pr\big(\mathbf{t}_n|\mathbf{x}_n, \alpha, (\theta_k)_{k=1}^K\big). \tag{1}$$

This modelling and objective are almost identical to the Latent Dirichlet Allocation. Readers are referred to Appendix C for the details of inference of $(\theta_k)_{k=1}^K$.

**Remark 1** *The LDA-based method presented in this section is a variant of the mini-batch based LDA in (Hoffman et al., 2010). Therefore, the proposed method also benefits from the convergence analysis in (Hoffman et al., 2010, Section 2.3), where the learning can be considered as a stochastic natural gradient algorithm (Sato, 2001).*

Note that the Dirichlet prior parameter $\alpha$ in Eq. (1) controls the *sparsity* of the inferred clusters (equivalent to topics in topic modelling) within an annotation set (equivalent to a document) of an instance. A small value of $\alpha$ promotes a sparse distribution, suggesting that each annotation set of an instance (or document) is likely to be generated from only a few clusters (or topics). Conversely, a large value of $\alpha$ encourages a more uniform distribution, implying that the annotation set (or document) is generated from a broader range of clusters (or topics). In human-annotated data, the inter-expert agreement is often high (Nowak & Rüger, 2010), making a larger value of $\alpha$ a preferred choice, as it reflects the consensus among experts and the presence of multiple, correlated clusters. In our empirical evaluation, we observe that a large value of $\alpha$ leads to the successful inference of multiple clusters, whereas a small value of $\alpha$ consistently resulted in a single, collapsed cluster.

One potential issue to the LDA-based clustering method is class-imbalanced. In this setting, under-represented clusters are harder to be estimated, potentially leading to cluster collapse or merging. To mitigate this, the Dirichlet prior parameter can be increased to encourage a more uniform prior distribution over clusters, forcing the model to learn all of the cluster equally, and hence, reducing the risk of collapsing or merging. In our experiments, we set $\alpha$ to a large value (e.g., $\alpha \geq 2$), which also minimises the effect of this issue.

## 3.2 Assignment of test-time human experts into clusters

At test time, the learnt LDA-based model (i.e., $(\theta_k^*)_{k=1}^K$) will be used to process a small *identified* set of data annotated by that human expert, denoted as $\mathcal{I}_m = \{(\mathbf{x}_j^{(m)}, t_j^{(m)})\}_{j=1}^{N_m}$. This corresponds to the on-boarding process of a new human expert in order to infer the probability of cluster membership of that human. This is equivalent to finding the expected value of the mixture coefficient $\mathbf{u}$ of the graphical model shown in Fig. 4 (Appendix C) over all instances $\mathbf{x}_j^{(m)} \in \mathcal{I}_m$. This cluster assignment can be done by calculating either *(i)* the expectation of the posterior of the cluster coefficient, or *(ii)* the Dirichlet parameter $\alpha$ representing the belief of the cluster assignment $\mathbf{u}$. As shown in Appendix F, the former approach requires an additional approximation, potentially leading to higher error. Hence, we decide to follow the latter approach, which is to determine the parameter of the Dirichlet distribution that generates $\mathbf{u}$. Such a parameter can be obtained by maximising the likelihood on the observed identified data $\mathcal{I}_m$ as follows:

$$\alpha^{(m)} = \operatorname{argmax}_\alpha \sum_{j=1}^{N_m} \ln \Pr\left(t_j^{(m)} \,\Big|\, \mathbf{x}_j^{(m)}, \alpha, (\theta_k^*)_{k=1}^K\right). \tag{2}$$

Please refer to Appendix D.3.2 for the detailed derivation to optimise for $\alpha$.

We then utilise the distribution of cluster membership $\Pr(\mathbf{u}|\alpha^{(m)})$ with $\alpha^{(m)}$ obtained in Eq. (2) and use its mean for a more interpretable representation of the assignment probability as follows:

$$\overline{\alpha}_k^{(m)} = \Pr(\mathsf{cluster}(\mathcal{I}_m) = k) \approx \alpha_k^{(m)}/\mathbf{1}^\top \alpha^{(m)}. \tag{3}$$

The probability vector of cluster assignment $\alpha^{(m)}$ in Eq. (3) for each of the $M$ test-time human experts can be stacked column-wise to form a matrix $\Omega = \begin{bmatrix} \overline{\alpha}^{(1)} & \overline{\alpha}^{(2)} & \dots & \overline{\alpha}^{(M)} \end{bmatrix}$ that represents the clusters of human experts. When performing downstream tasks, such as deferring to the cluster indexed by $k$ (see Section 3.3), one can simply sample a random human expert with a probability vector proportional to the $k$-th row of the cluster matrix $\Omega$. The whole procedure of the proposed method is summarised in Algorithm 1 (see Appendix I).

## 3.3 Learning to defer with clusters of human experts

We consider an L2D setting consisting of an annotated dataset: $\mathcal{D} = \{(\mathbf{x}_n, (\hat{\mathbf{y}}_n^{(m)})_{m=1}^M, \mathbf{y}_n)\}_{n=1}^{N'}$, where $\mathbf{x}_n \in \mathcal{X}$ is an input sample, $\hat{\mathbf{y}}_n^{(m)} \in \mathcal{Y} = \{1, \dots, C\}$ is the annotation made by a human expert indexed by $m \in \{1, \dots, M\}$ and $y_n \in \mathcal{Y}$ is the ground truth label. These human experts have been assigned to one of the $K$ clusters with their probability of cluster assignment defined in the matrix $\Omega$ (see Eq. (3) and the explanation after that). The aim is to learn an L2D system that can make an automated prediction, or defer to one cluster and have one random human expert in that cluster make the final decision. Fig. 1 provides a visual illustration for this application.

In this view, the desired system is analogous to a two-level hierarchical mixture of experts (Jordan & Jacobs, 1994), in which the first level corresponds to each of the clusters, while the second level consists of human experts in each of those clusters. In addition, due to the presence of the classifier (the only learnable expert), the given cluster assignment probability matrix $\Omega$ has to be expanded from $K$ to $K+1$ clusters by adding the classifier into an additional and separated cluster as follows:

$$\Omega = \begin{bmatrix} \overline{\alpha}^{(1)} & \overline{\alpha}^{(2)} & \dots & \overline{\alpha}^{(M)} & \mathbf{0} \\ 0 & 0 & \dots & 0 & 1 \end{bmatrix}, \tag{4}$$

where the last column denotes the probability of cluster assignment for the classifier. We then extend the *Probabilistic L2D* framework (Nguyen et al., 2025) to a two-level hierarchical mixture of experts to model the data generation process for $K+1$ clusters of experts (human and classifier) as follows:

1. draw a sample: $\mathbf{x} \sim \Pr(\mathbf{x})$,
2. draw a cluster index: $\mathbf{v} \sim \operatorname{Categorical}(\mathbf{v}|g(\mathbf{x}; \gamma))$,
3. draw an expert's index from that cluster: $\mathbf{w} \sim \operatorname{Categorical}\left(\mathbf{w} \,\big|\, \Omega^\top \vec{\mathbf{e}}_v/\mathbf{1}^\top \Omega^\top \vec{\mathbf{e}}_v\right)$,
4. draw an annotation made by that expert: $\hat{\mathbf{y}}^{(w)} \sim \operatorname{Categorical}(\hat{\mathbf{y}}|f(\mathbf{x}; \phi_w))$,
5. draw a ground truth label: $\mathbf{y} \sim \operatorname{Categorical}\left(\mathbf{y}|\hat{\mathbf{y}}^{(w)}\right)$,

where: $g(.;\gamma) : \mathcal{X} \rightarrow \{1, \ldots, K+1\}$ is the gating model, $\vec{e}_j$ is the $j$-th unit coordinate vector, and $f(.;\phi_m) : \mathcal{X} \rightarrow \Delta_{C-1}$ is a classification model representing an expert (including the classifier) indexed by $m \in \{1, \ldots, M+1\}$. Note that the function representing each human expert, denoted as $f(.;\phi_m), \forall m \in \{1, \ldots, M\}$, is simply a mapping table that returns an annotation $\hat{\mathbf{y}}^{(m)} = f(\mathbf{x};\phi_m)$ given an input $\mathbf{x}$. This data modelling approach can visually be illustrated in the graphical model shown in Fig. 5 (see Appendix G).

The proposed method, named as L2D-Clusters, does not require all human experts to annotate the same set of training samples, but each cluster must have at least one expert (not necessary the same expert) with probability greater than zero to annotate each sample. In fact, our proposed framework can be extended to handle "missing annotations" using techniques similar to Probabilistic-L2D (Nguyen et al., 2025), where unobserved annotations are treated as latent random variables. To simplify the analysis, clearly isolate our contribution (cluster-based deferral), and ensure fair comparison with existing baselines (e.g., most of which assume complete annotations), we assume that all human experts annotate the same $N'$ samples to simplify the formulation. Readers are referred to Appendix G for the detailed formulation and parameter inference of the proposed L2D-Clusters.

**Constrain workload assignment for the classifier**   Without constraining the workload assignment, L2D-Clusters will be biased to defer all the queries to a cluster containing high-performing human experts, ignoring the classifier. Although such a biased workload distribution results in higher overall prediction accuracy, it tends to overload one particular cluster, and diminishes the role of the classifier in the L2D system, resulting in a cost-ineffective solution. Hence, we follow a similar strategy as *Probabilistic-L2D* (Nguyen et al., 2025) to impose a constraint to control the workload distributed to the cluster of the classifier (see Appendix H for further details).

Naively training L2D-Clusters with the standard EM algorithm is computationally prohibitive for large-scale datasets, as each iteration requires processing the entire training set. To address this, we adopt the online EM algorithm (Cappé & Moulines, 2009), which enables mini-batch updates and thus scales efficiently. Online EM maintains an exponential moving average to approximate the complete-data likelihood, which can be implemented as momentum in gradient descent. Accordingly, we train L2D-Clusters using SGD with a high momentum. This approach inherits the convergence guarantees to stationary points established in (Cappé & Moulines, 2009).

## 4    ONBOARDING L2D BENCHMARK

Most existing studies in L2D adopt a human-specific modelling approach, and consequently, the associated benchmarks typically rely on large *context* datasets in which each expert provides a substantial number of annotations. While these benchmarks are widely used, they fail to reflect practical scenarios such as expert onboarding, where annotators typically contribute only a limited number of samples. Although recent works have explored settings with incomplete or "missing" annotations (Hemmer et al., 2023; Nguyen et al., 2025), their training procedures still assume a sufficiently large volume of data per expert and typically involve only a small number of experts (usually less than 5). In contrast, real-world datasets often comprise many experts, each contributing only a few annotations. Furthermore, most synthetic expert benchmarks in L2D simulate annotation errors using class-dependent but instance-independent label noise. This modelling assumption is overly simplistic and does not align with empirical observations, as real-world label noise is frequently both class- and instance-dependent (Carneiro, 2024).

We, therefore, propose a new benchmark that more accurately reflects real-world annotation dynamics. Specifically, synthetic experts are simulated to make annotation errors under a class- and instance-dependent noise model (Xia et al., 2020). The training set is constrained to a small number of annotations per expert (e.g., fewer than 200 samples), emulating realistic onboarding conditions. Additionally, to capture the high inter-annotator agreement observed among human experts (Nowak & Rüger, 2010), we simulate multiple synthetic experts sharing identical annotation noise rate with different random seeds, thereby modelling consistent annotation behaviour within expert groups.

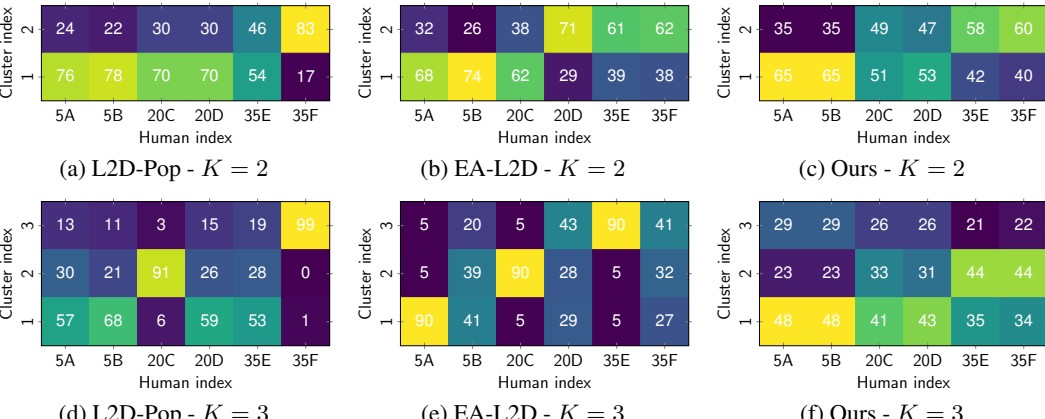

(a) L2D-Pop - $K = 2$      (b) EA-L2D - $K = 2$      (c) Ours - $K = 2$

(d) L2D-Pop - $K = 3$      (e) EA-L2D - $K = 3$      (f) Ours - $K = 3$

Figure 2: Clustering results on the simulation of 6 synthetic experts on Cifar-100: the numbers in each column represents the probability of clustering each human expert (x axis) into clusters (y axis) evaluated on a context set of $N' = 100$ samples annotated by the corresponding human expert. Each human expert is simulated to annotate with 5%, 20% and 35% of mistakes at different random seeds, following the instance-dependent label noise. Ideally, each pair will be grouped into the same cluster. Note that the heatmap colour is relatively scaled to enhance the visualisation.

## 5 EXPERIMENTS

We empirically evaluate the performance of the clustering method presented in Section 3.1 and the L2D-Clusters proposed in Section 3.3. All the results reported in this section are obtained from the checkpoint at the last iteration in each training.

### 5.1 ANALYSIS OF THE CLUSTERING ON ANONYMOUSLY-ANNOTATED DATA

**Baselines** We consider two baselines: L2D-Pop (Tailor et al., 2024) and EA-L2D (Strong et al., 2025). These two methods obtain human representation either via learning, as in L2D-Pop, or hand-crafting, as in EA-L2D. The set of representations of all human experts extracted from these two baselines are then clustered, and compare with our method presented in Section 3. Specifically, the whole model of L2D-Pop is trained on the *context* (i.e., with ground-truth labels) data of all available human experts to extract the representation of those experts. For EA-L2D, the statistics of each *context* set associated with each human expert is summarised via counting, and the mean and variance of the obtained Beta posterior are used to represent each human expert. The set of representations extracted from each baseline is then clustered by the Fuzzy C-Means algorithm (Dunn, 1973; Bezdek, 2013) – a variant of the K-Means algorithm. Note that other clustering algorithms, such as Gaussian mixture models, may also be used. However, due to the high-dimensional nature of the representation obtained from these baselines (e.g., in the range of hundreds) coupled with a small number of samples (e.g., in our setting, the number of human experts is less than 10), learning full covariance matrices of a Gaussian mixture model is challenging.

**Experiment setting** We simulate three pairs of synthetic annotators on CIFAR-100, each introducing 5%, 20%, and 35% annotation errors under class- and instance-dependent label noise. These annotators are denoted as 5A, 5B, 20C, 20D, 35E, and 35F, where the letter indicates different random seeds. Our method trains the clustering model on the full training set with anonymous annotations. During evaluation, the model assigns clusters based on the *identified* annotations of each expert (without ground truth). In contrast, L2D-Pop and EA-L2D use *context* data to extract expert representations, which are then clustered using Fuzzy C-Means. Further details are in Appendix J.

**Qualitative results** The clustering heatmap in Fig. 2 shows that both the baselines, including L2D-Pop and EA-L2D, fail to group each pair of synthetic human experts into the same cluster, while our proposed clustering method consistently find the annotation pattern of each pair of synthetic experts. For $K = 2$ clusters (top row of Fig. 2), L2D-Pop (Fig. 2a) fails to group the synthetic expert 35E and 35D together, while EA-L2D (Fig. 2b) misclusters experts 20C and 20D. In contrast, our method (Fig. 2c) correctly groups experts 5A-5B, and 35E-35F, while inferring 20C and 20D as a mixture of these two groups. For $K = 3$ clusters (bottom row of Fig. 2), neither baseline achieves consistent grouping of expert pairs. This outcome is expected: L2D-Pop exhibits poor performance

in low-data regimes (here, $N' = 100$ samples) due to its reliance on data abundance, while EA-L2D assumes class-dependent instance-independent annotation and hence, cannot work with the more realistic instance-dependent label noise. In contrast, our method (Fig. 2f), although not grouping expert pairs into perfectly separated clusters, reveals a smooth transition of the cluster assignment probabilities across pairs. Specifically, Cluster 1 demonstrates the highest association with the synthetic pair 5A-5B, followed by a gradual decrease in association with pairs 20C-20D and 35E-35F. A similar trend is observed for the remaining clusters. This reflects substantial annotation overlap (approximate 70% agreement), which introduces ambiguity in forming distinct clusters. Nevertheless, within each cluster, paired experts exhibit nearly identical probabilities, demonstrating our model's ability to capture annotation similarity – a property absent in L2D-Pop and EA-L2D. Appendix K provides additional results with higher number of clusters.

**Quantitative results** Given the methodological differences between approaches (e.g., baseline methods are distance-based, whereas our approach is distributional and likelihood-based), our quantitative evaluation relies on ground-truth-based metrics, where expert pairs (5A, 5B), (20C, 20D), and (35E, 35F) are treated as reference clusters. Performance is then measured using the *Adjusted Rand Index* (Hubert & Arabie, 1985) and *Adjusted Normalised Mutual Information* (Vinh et al., 2009). As shown in Table 1, our proposed method consistently outperforms the two baselines, despite relying solely on the

Table 1: The adjusted Rand index (ARI) and adjusted normalised mutual information (ANMI) of the expert clustering.

| Method | ARI ($\uparrow$) | ANMI ($\uparrow$) |
|---|---|---|
| *K = 3 clusters* | | |
| L2D-Pop | -0.0606 | -0.0870 |
| AE-L2D | 0.4444 | 0.5024 |
| **Ours** | 0.4444 | 0.6154 |

*identified* set of human annotations (without ground-truth) rather than the *context* annotation set (with ground-truth) used by the baselines. This improvement is primarily attributed to the robust identification of expert pairs, even though these pairs of experts are not perfectly assigned to separated clusters.

**Number of clusters** can be selected through various approaches, such as non-parametric learning with *Dirichlet process* (Blei & Jordan, 2004) or hyper-parameter optimisation. For simplicity, we follow the latter approach by evaluating the perplexity with different number of clusters $K$ on a hold-out set of anonymously-annotated data and selecting $K$ that minimises perplexity. However, as perplexity is known to exhibit a tendency toward overfitting, whereby increasing $K$ often leads to smaller perplexity values regardless of model quality (Chang et al., 2009). To mitigate this issue, we adopt a heuristic metric known as the *rate of perplexity change* (Zhao et al., 2015), which identifies the smallest value of $K$ beyond which further relative reductions in perplexity become marginal. For example, in the simulation on Cifar-100 with six synthetic human experts, the optimal number of clusters is determined to be $K = 3$, corresponding to the point of maximum rate of perplexity change on the hold-out set (see Fig. 7a). Please refer to Appendix L for further details.

**Sensitive analysis on the number of clusters** is performed on Cifar-100 with 100 experts and identified set size of 100 samples and reported as the area under coverage-accuracy curve (AUC-AC) for different values of $K$ in Table 2*(left)*. Increasing the number of clusters $K$ can indeed allow the model to capture finer-grained annotation patterns within the anonymously annotated dataset. However, setting $K$ excessively high introduces redundant clusters whose probabilities remain near zero across all annotation sets, effectively making them "unused". Empirically, we observe that the dataset exhibits an intrinsic cluster structure with $K_{\text{intrinsic}} = 4$. Beyond this point, additional clusters do not contribute with meaningful information, leading to a plateau in performance.

Despite that, fixing $K = K_{\text{intrinsic}}$ is not always optimal, as it may risk overfitting to the training data. To address this, our paper adopts a principled heuristic based on the *rate of perplexity change* (Zhao et al., 2015) evaluated on a held-out set (details in Appendix L). This metric provides a more robust criterion for selecting $K$. Importantly, our experiments show that performance remains stable across a range of $K$, indicating that the model is relatively insensitive to this hyperparameter.

**Sensitive analysis on Dirichlet prior** is also performed to quantify clustering stability by computing the average pairwise cosine dissimilarity of the cluster assignment distributions defined in Eq. (3). This metric captures how dissimilar the cluster membership probabilities are between every pair of human experts, in which a larger value represents a more diverse clustering. In addition, we report the AUC-AC score in Table 2*(right)*, which reflects the overall system performance across coverage–accuracy trade-offs. Further increasing $\alpha$ leads to a plateau in both the pairwise cosine

Table 2: Sensitive analysis on *(left)* number of clusters $K$ in which the AUC-AC results are stable at various $K$, and *(right)* the Dirichlet prior's parameter $\alpha$ in which a larger value of $\alpha$ promotes a more diverse clustering (higher cosine dissimilarity of cluster assignment probability pair).

| № clusters | AUC-AC |
|---|---|
| 2 | $73.72 \pm 1.01$ |
| 3 | $74.27 \pm 0.99$ |
| 4 | $74.00 \pm 0.99$ |
| 5 | $74.23 \pm 0.99$ |
| 6 | $74.04 \pm 1.00$ |
| 7 | $74.19 \pm 0.99$ |
| 8 | $74.17 \pm 1.00$ |
| 9 | $74.05 \pm 0.97$ |
| 10 | $74.07 \pm 1.00$ |

| $\alpha$ | Pair-wise cosine dissimilarity ($\uparrow$) | AUC-AC ($\times 100 \uparrow$) |
|---|---|---|
| 0.1 | 0.0002 | $72.90 \pm 2.00$ |
| 0.9 | 0.0009 | $73.10 \pm 1.77$ |
| 2.0 | 0.0377 | $74.98 \pm 1.32$ |
| 10 | 0.0406 | $75.07 \pm 1.06$ |
| 50 | 0.0422 | $75.04 \pm 1.25$ |

similarity and the performance (i.e., AUC-AC). This occurs because $\alpha$ governs the prior cluster distribution $\Pr(\mathbf{u}|\alpha)$, whereas the posterior of cluster assignment for the human expert indexed by $m$ (see Section 3.2), denoted as $\Pr(\mathbf{u}|\alpha, \mathcal{I}_m)$, depends on the observed identified set $\mathcal{I}_m$. Thus, while a larger $\alpha$ promotes uniformity of cluster distribution, the posterior $\Pr(\mathbf{u}|\alpha, \mathcal{I}_m)$ for the human expert $m$ does not need to be uniform. If an expert's annotations strongly favour certain clusters, the posterior will remain skewed toward those clusters despite the prior bias, which results in the plateau in both the pairwise cosine dissimilarity and AUC-AC.

## 5.2 LEARNING TO DEFER WITH SMALL CONTEXT SETS

We employ the clustering results obtained in Section 5.1 to train an L2D-Clusters system presented in Section 3.3, and use the benchmark in Section 4 to evaluate. Our evaluation focuses on the trade-off between coverage (i.e., inversely proportional to operational cost) and accuracy (i.e., performance) via the *coverage - accuracy* curves. *Coverage* computes the percentage of test samples assigned and predicted by the classifier, while *accuracy* measures the final prediction.

We report the two variants of our method: one with the "soft" clustering probability, and the other is its "hard" version, where each human is assigned to only a single cluster (e.g., the probability vector is rounded to be a one-hot vector). Here, we use $K = 2$ when training our clustering model as it shows the optimal perplexity on the hold-out validation samples. In addition, as the proposed L2D-Clusters is stochastic in terms of selecting a human expert in a cluster to make decisions, we, therefore, calculate the average prediction accuracy weighted by the assignment probability of each human expert belonging to the selected cluster: $\text{accuracy}_{\text{L2D−Clusters}}(\mathbf{z} = k) = \sum_{m=1}^{M+1} \Omega_{km} \mathbb{1}_{(\mathbf{t}^{(m)}=\mathbf{y})}$, where: $\Omega_{km}$ is the probability of expert indexed by $m$ in (the cluster indexed by $k$ can be selected to make decisions (see Eqs. (3) and (4)) and $\mathbb{1}$ is the indicator function. This additional step is to ensure that our comparison is fair to other methods.

**Experiment setting** We generate 3 pairs of synthetic experts on Cifar-100 similar to the ones in Section 5.1. A similar simulation is also applied on the dataset dopanim. For the real-world dataset Chaoyang, we use the 3 human pathologists provided in the dataset.

**Baselines** We consider two types of baselines: *(i)* the conventional human-specific L2D, including one-stage (Mozannar et al., 2023; Verma et al., 2023) and two-stage L2D methods (Madras et al., 2018; Mao et al., 2023; Nguyen et al., 2025), and *(ii)* methods that can handle the dynamic availability of human experts between training and testing, including LECODU (Zhang et al., 2024), L2D-Pop (Tailor et al., 2024). When there are enough training samples: (i.e., $N'$ is sufficiently large), the human-specific L2D upper-bounds the performance of the proposed L2D-Clusters. Our interest is more in the low-data regime (e.g., small *context* sets with ground-truth labels). For a fair comparison, we train LECODU directly on anonymous dataset without having the learning to complement part, and set the deferral to only one random expert. Hence, LECODU in this setting is equivalent to L2D-Clusters with all human experts uniformly assigned to one cluster. The other baseline, L2D-Pop, however, can only defer to one specific human at a time. To adapt to the setting of multiple available human experts at test time, for each sample, we calculate the deferral probabilities of all available human experts, and consider the mode consisting the human with the maximal

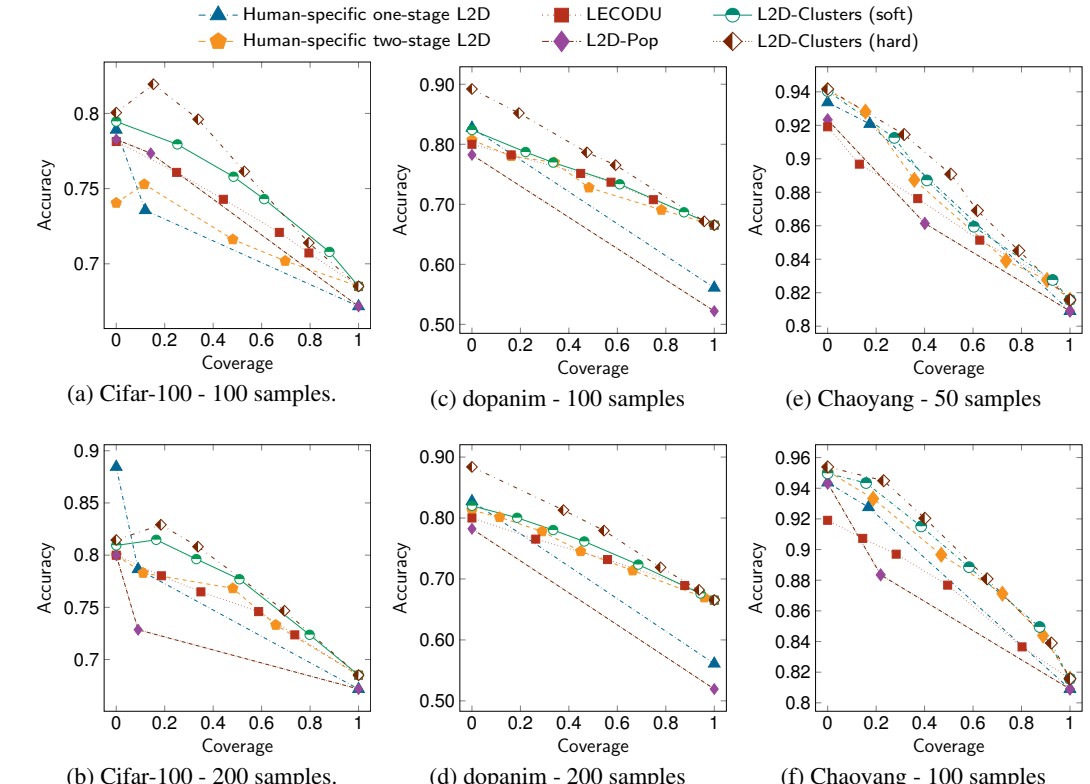

Figure 3: Comparison of coverage - accuracy curves between different the standard L2D methods (i.e., single-stage and two-stage), L2D-Pop, LECODU and our proposed L2D-Clusters on a variety of datasets, each with different number of identified/context samples $N'$.

Table 3: The area under the coverage - accuracy curve ($\times 100$) of many L2D methods at different number of context/identified set sizes with mean and standard deviation.

| Context size $N'$ | Cifar-100 (6 experts) | | dopanim | | Chaoyang | |
|---|---|---|---|---|---|---|
| | 100 | 200 | 100 | 200 | 50 | 100 |
| One-stage | $71.07 \pm 2.34$ | $73.87 \pm 1.97$ | $69.42 \pm 1.04$ | $69.42 \pm 1.05$ | $87.57 \pm 0.96$ | $87.97 \pm 0.94$ |
| Two-stage | $71.80 \pm 1.55$ | $75.02 \pm 1.53$ | $73.35 \pm 1.00$ | $74.03 \pm 1.00$ | $87.44 \pm 0.85$ | $89.31 \pm 0.84$ |
| LECODU | $73.60 \pm 1.98$ | $74.81 \pm 1.72$ | $74.00 \pm 1.00$ | $73.67 \pm 0.99$ | $86.46 \pm 0.85$ | $85.57 \pm 0.85$ |
| L2D-Pop | $73.05 \pm 2.65$ | $70.59 \pm 2.17$ | $65.23 \pm 1.52$ | $65.08 \pm 1.52$ | $85.81 \pm 1.73$ | $86.09 \pm 1.70$ |
| L2D-Clusters (soft) | $75.07 \pm 1.06$ | $76.68 \pm 1.05$ | $74.61 \pm 0.72$ | $75.09 \pm 0.72$ | $87.81 \pm 0.85$ | $89.69 \pm 0.83$ |
| L2D-Clusters (hard) | $\mathbf{76.17} \pm 1.50$ | $\mathbf{77.31} \pm 1.48$ | $\mathbf{78.12} \pm 0.98$ | $\mathbf{78.20} \pm 0.97$ | $\mathbf{88.51} \pm 0.87$ | $\mathbf{90.09} \pm 0.85$ |

deferral probability. In addition, the one-stage L2D (Verma & Nalisnick, 2022) and L2D-Pop (Tailor et al., 2024) do not provide a mechanism to control the coverage. Hence, we can only report a single point beside the two extreme points of coverages of zero and one. Note that, we do not consider EA-L2D as a baseline because EA-L2D always assumes a fixed prediction of each human expert over all samples (Strong et al., 2025, Eq. (4)), resulting in extremely poor human performance, enforcing the rejector of EA-L2D to defer to the classifier all the time (i.e., zero coverage).

**Results** Fig. 3 shows the *coverage - accuracy* curves of the four baselines and L2D-Clusters on various datasets, each at different numbers of annotator-identified samples $N'$. In general, both the human-specific baselines (one-stage and two-stage) and L2D-Pop overfit to small training *context* sets, resulting in low performance. LECODU with its nature of random selection, performs most likely overlaps with the line connecting the average performance over different coverage values. In contrast, the proposed L2D-Clusters (both soft and hard) outperforms these baselines, demonstrating its capability of dealing with real-world annotated datasets when only a small amount of training samples are associated with the (pseudo) identities of human experts. Note that although our proposed clustering method presented in Section 3 can handle small $N'$ in its inference, further reducing $N'$ makes the training of L2D methods difficult due to the insufficient number of training samples. For example, Cifar-100 has 100 classes, and hence, theoretically needs at least $N' = 100$ samples to train, while for Chaoyang, $N'$ can be slightly smaller. For larger $N'$ (see Appendix M),

the performance of L2D-Clusters also improves, but becomes worse than the human-specific one-stage and two-stage L2D methods. This is expected because at larger $N'$, the traditional one-stage and two-stage L2D baselines have enough information about each specific human expert, and hence, upper-bounds the performance of our L2D-Clusters.

We also report the quantitative results as the area under the coverage - accuracy curve of all the baselines and our proposed L2D-Clusters. The results in Table 3 show that our proposed L2D-Clusters out-performs all the baselines, in which the hard version of L2D-Clusters has the highest performance across all the practical settings when the size of *identified/context* set, $N'$, is small.

**Large number of human experts**  We also study the effectiveness of our proposed framework with large number of experts. Specifically, we simulate a total of 100 synthetic experts on Cifar-100, each has an error rate varying from 5% to 50%. The results shown in Fig. 9 are consistent with our previous results, in which our proposed framework, including both LDA-based clustering and L2D-Clusters, outperforms the baselines and state-of-the-art methods in the onboarding setting (e.g., small context sets). Please refer to Appendix N for further details.

## 6 RELATED WORK

As our focus is on the context of L2D, this section is, therefore, dedicated to review related study in L2D. Readers are referred to Appendix B for the extension of the related work, including clustering crowdsourced workers as well as consensus labels from crowd.

**Learning to defer**  has recently attracted research interest due to its inherent capability to make autonomous decisions or selectively defer to a human expert when lacking sufficient confidence (Madras et al., 2018; Mozannar & Sontag, 2020; Keswani et al., 2021; Verma & Nalisnick, 2022; Verma et al., 2023; Mozannar et al., 2023; Nguyen et al., 2025). However, a prevalent characteristic among most existing L2D methodologies is their reliance on a human-specific deferral strategy, which often fails to account for the dynamic availability of human experts encountered in real-world deployment scenarios, distinct from development environments. To our best knowledge, only *Learning to Defer to a Population* (L2D-Pop) (Tailor et al., 2024) and *Expert-Agnostic Learning to Defer* (EA-L2D) (Strong et al., 2025) circumvent this limitation by learning the representation of each human expert following a data-driven (L2D-Pop) or hand-crafted approach (EA-L2D), and training L2D models conditioned on such representation. Although these human-agnostic approaches increase the flexibility of L2D, both require a sufficient *context* set containing not only the annotations by that expert, but also the ground truth associated with each sample. In contrast, our clustering model presented in Section 3.1 can be trained on anonymous annotation data which is more accessible in practice. In addition, our proposed clustering approach does not need a *context* dataset (with ground truth) per each human expert, but only an *identified* dataset (without ground truth) to assign that human expert into one of the clusters. Furthermore, the L2D-Clusters approach developed in Section 3.3 introduces a cluster-based deferral mechanism and chooses one expert within the group being selected for the final decision. Such deferral mechanism flexibility enables L2D-Clusters to readily adapt to diverse real-world scenarios (e.g., absence, retirement or on-boarding), while simultaneously demonstrating performance comparable to traditional human-specific L2D methods.

## 7 CONCLUSION

This paper proposes an LDA-based probabilistic method to uncover annotation patterns among anonymous annotators, enabling efficient onboarding by clustering new members with minimal annotations. We also extend L2D from expert-specific to cluster-based models that defer to groups of experts. To evaluate this, we introduce a benchmark simulating onboarding with few annotations per expert under instance-dependent label noise. Our LDA-based clustering aligns well with known annotator performance similarities. On the onboarding benchmark, L2D-Clusters combined with our clustering method consistently outperforms expert-specific and expert-agnostic L2D approaches. However, the method faces challenges such as slow convergence due to EM's nature and cluster collapse from unbalanced data. Future work should integrate domain knowledge and advanced techniques, such as Dirichlet processes (Ferguson, 1973; Blei & Jordan, 2004), to learn more robust models.

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

## A    DISCLOSURE OF LARGE LANGUAGE MODEL USAGE

Parts of the text in this paper were originally written by the authors and revised by large language models to improve clarity. The origin of the paper, including conceptualisation, formulation, experiments and conclusion, were carried out by the authors themselves.

## B    EXTENDED RELATED WORK

**Clustering crowdsource annotators methods**    aim to group human annotators based on the similarity of their annotation patterns. Early approaches in this domain, such as *crowd-parting* (Kairam & Heer, 2016), simply measured the agreement and disagreement on shared annotated samples to quantify the similarity of annotators. Subsequent methods have employed more sophisticated statistical techniques, including Cohen's kappa and Krippendorff's alpha coefficients, to assess inter-annotator agreement and correlation (Wich et al., 2020; Basile, 2020; Lo & Basile, 2023). These methods, whether based on simple agreement counts or statistical coefficients, rely on overlapping set of samples, restricting their applicability in more general scenarios, such as distinct, non-overlapping subsets of data. More sophisticated modelling approaches based on probabilistic graphical models have also been proposed to address more complex annotation settings and model annotator reliability or bias (Nguyen et al., 2019; Traganitis & Giannakis, 2022). These models, however, assume a fixed set of experts, typically requiring retraining or complex inference procedures upon changes to the expert pool. Another line of research focuses on directly learning representations for individual human experts, which can then be used for clustering (Deng et al., 2023; Tailor et al., 2024). A key limitation of these representation-learning methods is the need of annotator-identified training data. In contrast to these existing approaches, our clustering method, detailed in Section 3, operates independently of the annotator-identified data during training. Furthermore, it can effectively utilise any set of annotated samples, irrespective of overlapping set of samples or the number of annotators per sample. The trained model can subsequently be used to cluster newly on-boarded annotators directly, without the need of re-training. Such capabilities of our proposed method offers enhanced flexibility and broader applicability, particularly in scenarios involving anonymously annotated datasets.

**Consensus labels from crowd**    is another related research topic aiming to elicit ground truth labels from manual annotations. In particular, these studies require access to identified annotations of human experts coupled with pre-trained classifiers to produce consensus labels (Showalter et al., 2024; Kelly et al., 2025). This is distinctively different from our study, in which we employ anonymous annotation data to learn a model to cluster human experts.

## C    DETAILS OF THE PARAMETER INFERENCE FOR THE LDA MODEL

The whole data generation process presented in Section 3 can also be illustrated via the graphical model shown in Fig. 4.

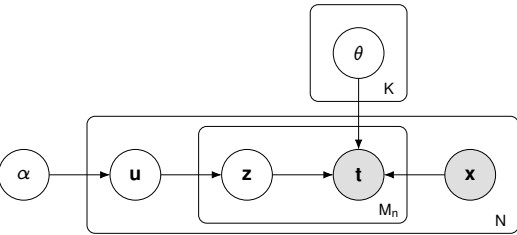

Figure 4: The graphical model describing the data generation of anonymous annotation $\mathbf{t}|\mathbf{x}$, which is generated from one of $K$ clusters, parameterised by $(\theta_k)_{k=1}^{K}$, with $\alpha$ being a Dirichlet parameter, $\mathbf{u}$ being the random variable representing the probability of selecting a cluster, and $\mathbf{z}$ being another random variable denoting the cluster selected.

To infer the parameters $(\theta_k)_{k=1}^K$ in Eq. (1), we follow the online learning approach for LDA in (Hoffman et al., 2010). In particular, the Expectation-Maximisation (EM) algorithm is employed to optimise a lower-bound of the objective function in (1). In that case, the "complete-data" log-likelihood in Eq. (1) for one sample is defined as follows:

$$Q\left(\mathbf{x}_n, \mathbf{t}_n, (\theta_k^{(\tau)})_{k=1}^K, (\theta_k)_{k=1}^K\right) = \mathbb{E}_{q(\mathbf{u}_n, \mathbf{z}_n | \mathbf{x}_n, \mathbf{t}_n, \alpha, (\theta_k^{(\tau)})_{k=1}^K)}\left[\ln \Pr\left(\mathbf{t}_n, \mathbf{z}_n, \mathbf{u}_n | \mathbf{x}_n, \alpha, (\theta_k)_{k=1}^K\right)\right]$$
$$= \mathbb{E}_{q(\mathbf{u}_n, \mathbf{z}_n | \mathbf{x}_n, \mathbf{t}_n, \alpha, (\theta_k^{(\tau)})_{k=1}^K)}\left[\ln \Pr\left(\mathbf{t}_n | \mathbf{x}_n, \mathbf{z}_n, (\theta_k)_{k=1}^K\right) + \ln \Pr(\mathbf{z}_n | \mathbf{u}_n) + \ln \Pr(\mathbf{u}_n | \alpha)\right],$$
(5)

where the superscript $\tau$ denotes the value at the $\tau$-th iteration.

**E-step** calculates the posterior of the latent variables $\mathbf{u}$ and $\mathbf{z}$ following the mean-field variational inference. In particular, the posterior of the two latent variables is assumed to be factorised as follows:

$$q(\mathbf{u}_n, \mathbf{z}_n | \mathbf{x}_n, \mathbf{t}_n, \alpha, (\theta_k^{(\tau)})_{k=1}^K) = \mathrm{Dir}(\mathbf{u}_n; \rho_n) \prod_{m=1}^{M_n} \mathrm{Categorical}(\mathbf{z}_{nm}; \zeta_{nm}). \tag{6}$$

The approximate posterior of the two latent random variables $\mathbf{u}$ and $\mathbf{z}$ can be obtained via the mean-field variational inference as follows:

$$(\rho_n^*, (\zeta_{nm}^*)_{m=1}^{M_n}) = \underset{(\rho_n, (\zeta_{nm})_{m=1}^M)}{\mathrm{argmin}} \mathrm{KL}\left[q(\mathbf{u}_n, \mathbf{z}_n | \rho_n, (\zeta_{nm})_{m=1}^{M_n}) \, \Big\| \, \Pr\left(\mathbf{u}_n, \mathbf{z}_n | \mathbf{x}_n, \mathbf{t}_n, \alpha, (\theta_k^{(\tau)})_{k=1}^K\right)\right]. \tag{7}$$

Readers are referred to Appendix D for the detailed derivation and optimisation of Eq. (7).

**M - step** maximises the "complete-data" log-likelihood in Eq. (5) w.r.t. $(\theta_k)_{k=1}^K$ as follows:

$$(\theta_k^{(\tau+1)})_{k=1}^K = \mathrm{argmax}_{(\theta_k)_{k=1}^K} {}^{1}/_{N} \sum_{n=1}^N Q\left(\mathbf{x}_n, \mathbf{t}_n, (\theta_k^{(\tau)})_{k=1}^K, (\theta_k)_{k=1}^K\right). \tag{8}$$

# D VARIATIONAL INFERENCE FOR THE LDA-BASED MODELLING

In this section, we provide the detailed derivation to approximate the posterior of the latent random variables presented in Appendix C. The objective function of the variational inference is rewritten as follows:

$$(\rho^*, \{\zeta_m^*\}_{m=1}^M) = \underset{(\rho, \{\zeta_m\}_{m=1}^M)}{\mathrm{argmin}} \mathrm{KL}\left[q(\mathbf{u}, \mathbf{z} | \rho, \zeta) \| \Pr\left(\mathbf{u}, \mathbf{z} | \mathbf{x}, \mathbf{t}, \theta^{(\tau)}, \alpha^{(\tau)}\right)\right]. \tag{7}$$

The KL divergence on the right-hand side in Eq. (7) can be expanded as follows:

$$\mathrm{KL}\left[q(\mathbf{u},\mathbf{z}|\rho,\zeta)\|\Pr\left(\mathbf{u},\mathbf{z}|\mathbf{x},\mathbf{t},\theta^{(\tau)},\alpha^{(\tau)}\right)\right]$$

$$= \mathbb{E}_q\left[\ln q(\mathbf{u},\mathbf{z}|\rho,\zeta) - \ln\Pr\left(\mathbf{u},\mathbf{z}|\mathbf{x},\mathbf{t},\theta^{(\tau)},\alpha^{(\tau)}\right)\right]$$

$$= \mathbb{E}_q\left[\ln q(\mathbf{u};\zeta) + \sum_{m=1}^{M}\ln q(\mathbf{z}_m;\zeta_m) - \ln\Pr\left(\mathbf{t}_m|\mathbf{x},\mathbf{z}_m,\theta^{(\tau)}\right) - \ln\Pr(\mathbf{z}_m|\mathbf{u})\right.$$

$$\left. - \ln\Pr(\mathbf{u}|\mathbf{x},\alpha)\right] + \mathrm{const.}$$

$$= \ln\Gamma\left(\sum_{k=1}^{K}\rho_k\right) - \sum_{k=1}^{K}\ln\Gamma(\rho_k) - \sum_{k=1}^{K}(\rho_k-1)\left[\psi\left(\sum_{j=1}^{K}\rho_j\right) - \psi(\rho_k)\right]$$

$$+ \sum_{m=1}^{M}\sum_{k=1}^{K}\zeta_{mk}\ln\zeta_{mk}$$

$$- \sum_{m=1}^{M}\sum_{k=1}^{K}\zeta_{mk}\sum_{c=1}^{C}\mathbf{t}_{mc}\ln f_c\left(\mathbf{x};\theta_k^{(\tau)}\right)$$

$$- \sum_{m=1}^{M}\sum_{k=1}^{K}\zeta_{mk}\left[\psi(\rho_k) - \psi\left(\sum_{j=1}^{K}\rho_j\right)\right]$$

$$+ \sum_{k=1}^{K}\ln\Gamma\left(\alpha_k^{(\tau)}\right) - \ln\Gamma\left(\sum_{k=1}^{K}\alpha_k^{(\tau)}\right) - \sum_{k=1}^{K}\left(\alpha_k^{(\tau)}-1\right)\left[\psi(\rho_k) - \psi\left(\sum_{j=1}^{K}\rho_j\right)\right], \quad (9)$$

where: $\Gamma(.)$ is the gamma function and $\psi(.)$ is the digamma function.

### D.1 Variational inference for the parameter of categorical distribution

We minimise the variational-free energy in Eq. (9) w.r.t. each $\zeta_m$. Note that $\zeta_m$ is a probability vector, meaning that:

$$\sum_{k=1}^{K}\zeta_{mk} = 1. \tag{10}$$

The Lagrangian for $\zeta_{mk}$ can then be written as follows:

$$\mathsf{L}_{\zeta_{mk}} = \zeta_{mk}\ln\zeta_{mk} - \zeta_{mk}\sum_{c=1}^{C}\mathbf{t}_{mc}\ln f_c\left(\mathbf{x};\theta_k^{(\tau)}\right) - \zeta_{mk}\left[\psi(\rho_k) - \psi\left(\sum_{j=1}^{K}\rho_j\right)\right] + \lambda_m\left(\sum_{j=1}^{K}\zeta_{mj}-1\right).$$
$$\tag{11}$$

Taking the derivative w.r.t. $\zeta_{mj}$ gives the following expression:

$$\frac{\mathrm{d}\mathsf{L}_{\zeta_{mj}}}{\mathrm{d}\zeta_{mj}} = \ln\zeta_{mj} + 1 - \sum_{c=1}^{C}\mathbf{t}_{mc}\ln f_c\left(\mathbf{x};\theta_j^{(\tau)}\right) - \psi(\rho_j) + \psi\left(\sum_{j=1}^{G}\rho_j\right) + \lambda_m. \tag{12}$$

Setting the derivative to zero yields the optimal solution as follows:

$$\boxed{\zeta_{mk} \propto \exp\left[\sum_{c=1}^{C}\mathbf{t}_{mc}\ln f_c\left(\mathbf{x};\theta_k^{(\tau)}\right) + \psi(\rho_j) - \psi\left(\sum_{j=1}^{K}\rho_j\right)\right].} \tag{13}$$

## D.2 Variational inference for the parameter of Dirichlet distribution

The variational-free energy for $\rho_j$ can be written as follows:

$$
\begin{aligned}
\mathsf{L}_\rho &= \ln \Gamma \left( \sum_{k=1}^{K} \rho_j \right) - \sum_{k=1}^{K} \ln \Gamma(\rho_k) - \sum_{k=1}^{K} (\rho_k - 1) \left[ \psi \left( \sum_{j=1}^{K} \rho_j \right) - \psi(\rho_k) \right] \\
&\quad - \sum_{m=1}^{M} \sum_{k=1}^{K} \zeta_{mk} \left[ \psi(\rho_k) - \psi \left( \sum_{j=1}^{K} \rho_j \right) \right] \\
&\quad + \sum_{k=1}^{K} (\alpha - 1) \left[ \psi \left( \sum_{j=1}^{K} \rho_j \right) - \psi(\rho_k) \right] \\
&= \sum_{k=1}^{K} \left[ \psi \left( \sum_{j=1}^{K} \rho_j \right) - \psi(\rho_k) \right] \left[ \alpha - \rho_k + \sum_{m=1}^{M} \zeta_{mk} \right] + \ln \Gamma \left( \sum_{j=1}^{G} \rho_j \right) - \ln \Gamma(\rho_k). \quad (14)
\end{aligned}
$$

Taking the derivative with respect to $\rho_k$ gives:

$$
\frac{\mathrm{d}\mathsf{L}_\rho}{\mathrm{d}\rho_k} = \left[ \psi' \left( \sum_{j=1}^{K} \rho_j \right) - \psi'(\rho_k) \right] \left[ \alpha - \rho_k + \sum_{m=1}^{M} \zeta_{mk} \right]. \quad (15)
$$

Setting the derivative to zero yields the optimal solution as follows:

$$
\boxed{\rho_k = \alpha + \sum_{m=1}^{M} \zeta_{mk}.} \quad (16)
$$

## D.3 Parameter estimation

Given the variational posterior of latent variables, $q$, one can perform the M-step to estimate the parameters of interest: $\theta$ and $\alpha$. The log-likelihood in this case is shown in Eq. (5), and hence, proportional to the lower-bound presented in Eq. (9). In particular, the log-likelihood in this case can be written as follows:

$$
\begin{aligned}
Q \left( \mathbf{x}_n, \mathbf{t}_n, (\theta_k^{(\tau)})_{k=1}^K, (\theta_k)_{k=1}^K \right) &= \mathbb{E}_{q(\mathbf{u}_n, \mathbf{z}_n | \mathbf{x}_n, \mathbf{t}_n, \alpha, (\theta_k^{(\tau)})_{k=1}^K)} \left[ \ln \Pr \left( \mathbf{t}_n, \mathbf{z}_n, \mathbf{u}_n | \mathbf{x}_n, \alpha, (\theta_k)_{k=1}^K \right) \right] \\
&= \mathbb{E}_{q(\mathbf{u}_n, \mathbf{z}_n | \mathbf{x}_n, \mathbf{t}_n, \alpha, (\theta_k^{(\tau)})_{k=1}^K)} \left[ \ln \Pr \left( \mathbf{t}_n | \mathbf{x}_n, \mathbf{z}_n, (\theta_k)_{k=1}^K \right) + \ln \Pr(\mathbf{z}_n | \mathbf{u}_n) + \ln \Pr(\mathbf{u}_n | \alpha) \right] \\
&= \sum_{m=1}^{M} \sum_{k=1}^{K} \zeta_{mk} \sum_{c=1}^{C} \mathbf{t}_{mc} \ln f_c \left( \mathbf{x}; \theta_k^{(\tau)} \right) \\
&\quad + \sum_{m=1}^{M} \sum_{k=1}^{K} \zeta_{mk} \left[ \psi(\rho_k) - \psi \left( \sum_{j=1}^{K} \rho_j \right) \right] \\
&\quad + \ln \Gamma \left( \sum_{k=1}^{K} \alpha_k^{(\tau)} \right) - \sum_{k=1}^{K} \ln \Gamma \left( \alpha_k^{(\tau)} \right) + \sum_{k=1}^{K} \left( \alpha_k^{(\tau)} - 1 \right) \left[ \psi(\rho_k) - \psi \left( \sum_{j=1}^{K} \rho_j \right) \right].
\end{aligned}
$$

$$(17)$$

### D.3.1 Estimate the parameter $\theta$

The term of the log-likelihood related to $\theta$ can be written as follows:

$$
Q_{[\theta]} = \frac{1}{N} \sum_{n=1}^{N} \sum_{m=1}^{M} \sum_{k=1}^{K} \zeta_{nmk} \sum_{c=1}^{C} \mathbf{t}_{nmc} \ln f_c \left( \mathbf{x}_n; \theta_k \right). \quad (18)
$$

It is then maximised to train the classifiers representing "pseudo-annotators".

### D.3.2   ESTIMATE THE PARAMETER $\alpha$

Similarly, the variational-free energy for $\alpha$ can be expressed as follows:

$$Q_{[\alpha]} = \ln \Gamma \left( \sum_{k=1}^{K} \alpha_k \right) - \sum_{k=1}^{K} \ln \Gamma \left( \alpha_k \right) + \sum_{k=1}^{K} \left( \alpha_k - 1 \right) \times \frac{1}{N} \sum_{n=1}^{N} \psi(\rho_{nk}) - \psi \left( \sum_{j=1}^{K} \rho_{nj} \right). \tag{19}$$

The gradient w.r.t. $\alpha_k$ can be written as follows:

$$\frac{\partial Q}{\partial \alpha_k} = \psi \left( \sum_{j=1}^{K} \alpha_j \right) - \psi(\alpha_k) + \frac{1}{N} \sum_{n=1}^{N} \psi(\rho_{nk}) - \psi \left( \sum_{j=1}^{K} \rho_{nj} \right). \tag{20}$$

The Hessian can then be written as follows:

$$\frac{\partial^2 Q}{\partial \alpha_k \partial \alpha_j} = \psi' \left( \sum_{j=1}^{K} \alpha_j \right) - \delta(k, j) \psi'(\alpha_k), \tag{21}$$

where: $\psi'(.)$ is the trigamma function.

The Newton - Raphson update step for $\boldsymbol{\alpha}$ in the online (i.e., minibatch) setting (Hoffman et al., 2010) can then be written as follows:

$$\boldsymbol{\alpha} \leftarrow \boldsymbol{\alpha} + (\tau_0 + \tau)^{-\kappa} \left[ \boldsymbol{\nabla}_{\boldsymbol{\alpha}}^2 \mathsf{L} \right]^{-1} \boldsymbol{\nabla}_{\boldsymbol{\alpha}} \mathsf{L}, \tag{22}$$

where: $\tau_0$ and $\kappa$ are hyper-parameters, and $\tau$ denotes the $\tau$-th iterations.

And since $\boldsymbol{\alpha}$ is vector with all elements being positive, we can use the result in Appendix E to update with a numerical stability guarantee:

$$\boldsymbol{\alpha} \leftarrow \boldsymbol{\alpha} \exp \left\{ \left[ \boldsymbol{\nabla}_{\boldsymbol{\alpha}}^2 Q \cdot \mathrm{diag}(\boldsymbol{\alpha}) + \mathrm{diag}(\boldsymbol{\nabla}_{\boldsymbol{\alpha}} Q) \right]^{-1} \cdot \boldsymbol{\nabla}_{\boldsymbol{\alpha}} Q \right\} \tag{23}$$

## E   NEWTON - RAPHSON METHOD IN LOG SCALE

In a minimisation problem, the update step of the Newton - Raphson method is as follows:

$$\mathbf{x}_{t+1} = \mathbf{x}_t - \left[ \boldsymbol{\nabla}_{\mathbf{x}}^2 h(\mathbf{x}) \right]^{-1} \bigg|_{\mathbf{x}=\mathbf{x}_t} \cdot \boldsymbol{\nabla}_{\mathbf{x}} h(\mathbf{x}) \bigg|_{\mathbf{x}=\mathbf{x}_t}, \tag{24}$$

where: $f : \mathbb{R}^K \to \mathbb{R}$.

When $\mathbf{x}$ is non-negative, then we can parameterise it as follows:

$$\mathbf{x} = \exp(\mathbf{z}). \tag{25}$$

Due to this change of variable, we have to calculate the gradient and Hessian w.r.t. the new variable $\mathbf{z}$. The gradient can be obtained via the chain rule as follows:

$$\boldsymbol{\nabla}_{\mathbf{z}} h(\mathbf{x}) = \boldsymbol{\nabla}_{\mathbf{z}} \mathbf{x} \cdot \boldsymbol{\nabla}_{\mathbf{x}} h(\mathbf{x}) = \mathrm{diag}(\mathbf{x}) \cdot \boldsymbol{\nabla}_{\mathbf{x}} h(\mathbf{x}), \tag{26}$$

where: $\mathrm{diag}(\mathbf{x})$ denotes the diagonal matrix formed by the vector $\mathbf{x}$.

For the Hessian, it may relate to *tensor dot product*. For simplicity, we use the result in (Skorski, 2019, Corollary 1) as follows:

$$\boldsymbol{\nabla}_{\mathbf{z}}^2 h(\mathbf{x}) = \left[ \boldsymbol{\nabla}_{\mathbf{z}} \mathbf{x} \right]^{\top} \cdot \boldsymbol{\nabla}_{\mathbf{x}}^2 h(\mathbf{x}) \cdot \boldsymbol{\nabla}_{\mathbf{z}} \mathbf{x} + \underbrace{\boldsymbol{\nabla}_{\mathbf{z}}^2 \mathbf{x} \cdot \boldsymbol{\nabla}_{\mathbf{x}} h(\mathbf{x})}_{\text{tensor product between a 3d tensor and a vector}}$$

$$= \mathrm{diag}(\mathbf{x}) \cdot \boldsymbol{\nabla}_{\mathbf{x}}^2 h(\mathbf{x}) \cdot \mathrm{diag}(\mathbf{x}) + \sum_{k=1}^{K} \frac{\partial h(\mathbf{x}}{\partial \mathbf{x}_k)} \frac{\partial \boldsymbol{\nabla}_{\mathbf{z}} \mathbf{x}}{\partial \mathbf{z}_k}$$

$$= \mathrm{diag}(\mathbf{x}) \cdot \boldsymbol{\nabla}_{\mathbf{x}}^2 h(\mathbf{x}) \cdot \mathrm{diag}(\mathbf{x}) + \sum_{k=1}^{K} \frac{\partial h(\mathbf{x}}{\partial \mathbf{x}_k)} \frac{\partial \mathrm{diag}(\mathbf{x})}{\partial \mathbf{z}_k}$$

$$= \mathrm{diag}(\mathbf{x}) \cdot \boldsymbol{\nabla}_{\mathbf{x}}^2 h(\mathbf{x}) \cdot \mathrm{diag}(\mathbf{x}) + \mathrm{diag}(\mathbf{x}) \cdot \mathrm{diag}(\boldsymbol{\nabla}_{\mathbf{x}} h(\mathbf{x})). \tag{27}$$

The Newton - Raphson update w.r.t. the new variable $\mathbf{z}$ can be written as follows:

$$
\begin{aligned}
\mathbf{z}_{t+1} &= \mathbf{z}_t - \left[\boldsymbol{\nabla}_{\mathbf{z}}^2 h(\mathbf{x})\right]^{-1} \boldsymbol{\nabla}_{\mathbf{z}} h(\mathbf{x}) \\
&= \mathbf{z}_t - \left\{\text{diag}(\mathbf{x}) \left[\boldsymbol{\nabla}_{\mathbf{x}}^2 h(\mathbf{x}) \cdot \text{diag}(\mathbf{x}) + \text{diag}(\boldsymbol{\nabla}_{\mathbf{x}} h(\mathbf{x}))\right]\right\}^{-1} \text{diag}(\mathbf{x}) \cdot \boldsymbol{\nabla}_{\mathbf{x}} h(\mathbf{x}) \\
&= \mathbf{z}_t - \left[\boldsymbol{\nabla}_{\mathbf{x}}^2 h(\mathbf{x}) \cdot \text{diag}(\mathbf{x}) + \text{diag}(\boldsymbol{\nabla}_{\mathbf{x}} h(\mathbf{x}))\right]^{-1} \cdot \boldsymbol{\nabla}_{\mathbf{x}} h(\mathbf{x}).
\end{aligned}
\tag{28}
$$

Hence, the update for the initial variable $\mathbf{x}$ can be written as:

$$
\mathbf{x}_{t+1} = \mathbf{x}_t \exp\left\{-\left[\boldsymbol{\nabla}_{\mathbf{x}}^2 h(\mathbf{x}) \cdot \text{diag}(\mathbf{x}) + \text{diag}(\boldsymbol{\nabla}_{\mathbf{x}} h(\mathbf{x}))\right]^{-1} \cdot \boldsymbol{\nabla}_{\mathbf{x}} h(\mathbf{x})\right\}.
\tag{29}
$$

# F  "AVERAGE" DISTRIBUTION WHEN INFERRING THE CLUSTER PROBABILITY

This section details the formulation to calculate the "average" distribution when inferring the cluster probability of a new human expert given the set of $N'$ samples annotated by that human expert. In particular, for each annotated sample $(\mathbf{x}_i, \mathbf{t}_i), \forall i \in \{1, \dots, N'$, one can apply the LDA model learnt in Section 3.2 to infer the variational posterior $q(\mathbf{u}_i|\mathbf{x}_i, \mathbf{t}_i; \rho_i) = \text{Dir}(\mathbf{u}; \rho_i)$. These variational distributions are then used to find the average distribution presented as follows:

$$
\overline{\rho}^* = \underset{\overline{\rho}}{\arg\min} \sum_{i=1}^{N'} \text{KL}\left[\text{Dir}(\mathbf{u}; \overline{\rho}) \| \text{Dir}(\mathbf{u}; \rho_i)\right].
\tag{30}
$$

Given that the Kullback - Leibler divergence between two Dirichlet distributions has a closed-form Eq. (33), the optimisation to find the average distribution can be written as follows:

$$
\begin{aligned}
\overline{\rho}^* = \underset{\overline{\rho}}{\arg\min} \sum_{i=1}^{N'} & \ln\Gamma\left(\sum_{k=1}^{K} \overline{\rho}_k\right) - \ln\Gamma\left(\sum_{k=1}^{K} \rho_{ik}\right) \\
& + \sum_{k=1}^{K} \ln\Gamma(\rho_{ik}) - \ln\Gamma(\overline{\rho}_k) + (\overline{\rho}_k - \rho_{ik})\left[\psi(\overline{\rho}_k) - \psi\left(\sum_{j=1}^{K} \overline{\rho}_k\right)\right].
\end{aligned}
\tag{31}
$$

It is simplified to:

$$
\begin{aligned}
\overline{\rho}^* = \underset{\overline{\rho}}{\arg\min}\ N' & \left[\ln\Gamma\left(\sum_{k=1}^{K} \overline{\rho}_k\right) - \sum_{k=1}^{K} \ln\Gamma(\overline{\rho}_k)\right] \\
& + \sum_{i=1}^{N'} \sum_{k=1}^{K} (\overline{\rho}_k - \rho_{ik})\left[\psi(\overline{\rho}_k) - \psi\left(\sum_{j=1}^{K} \overline{\rho}_k\right)\right].
\end{aligned}
\tag{32}
$$

This optimisation can be solved efficiently by employing the gradient descent with projection.

**Kullback - Leibler divergence between two Dirichlet distributions**  According to (Penny, 2001), the Kullback - Leibler divergence between two Dirichlet distributions has a closed-form as follows:

$$
\begin{aligned}
\text{KL}\left[\text{Dir}(\boldsymbol{\alpha}) \| \text{Dir}(\boldsymbol{\beta})\right] = & \ln\Gamma\left(\sum_{k=1}^{K} \boldsymbol{\alpha}\right) - \ln\Gamma\left(\sum_{k=1}^{K} \boldsymbol{\beta}_k\right) \\
& + \sum_{k=1}^{K} \ln\Gamma(\boldsymbol{\beta}_k) - \ln\Gamma(\boldsymbol{\alpha}_k) + (\boldsymbol{\alpha}_k - \boldsymbol{\beta}_k)\left[\psi(\boldsymbol{\alpha}_k) - \psi\left(\sum_{j=1}^{K} \boldsymbol{\alpha}_j\right)\right],
\end{aligned}
\tag{33}
$$

where: $\Gamma(.)$ is the gamma function and $\psi(.)$ is the digamma function.

## G  Parameter inference of L2D-Clusters

The data modelling of the L2D-Clusters presented in Section 3.3 can also be illustrated via the graphical model shown in Fig. 5.

The objective function of L2D-Clusters is to learn the gating model's parameter $\gamma$ and the classifier's parameter $\phi_{M+1}$ by maximising data likelihood on observed annotated data:

$$\max_{\gamma, \phi_{M+1}} \sum_{n=1}^{N'} \ln \Pr\left( \mathbf{y}_n, \prod_{m=1}^{M+1} \hat{\mathbf{y}}_n^{(m)} \middle| \mathbf{x}_n, \gamma, \Omega, \phi_{M+1} \right)$$

$$= \max_{\gamma, \phi_{M+1}} \sum_{n=1}^{N'} \ln \Pr\left( \mathbf{y}_n \middle| \mathbf{x}_n, \prod_{m=1}^{M+1} \hat{\mathbf{y}}_n^{(m)}, \gamma, \Omega \right) + \ln \Pr(\mathbf{y}_n | \mathbf{x}_n, \phi_{M+1}). \quad (34)$$

The first term in Eq. (34) learns the gating model $g(.; \gamma)$, while the second term learns the classifier $f(.; \phi_{M+1})$. We, therefore, train a classifier on ground truth data, and consider such a trained classifier as a fixed expert.

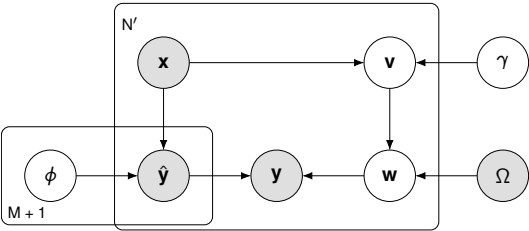

Figure 5: The graphical model of the data generation process of learning to defer to multiple clusters of human experts (L2D-Clusters) is modelled as a two-level hierarchical mixture of experts, in which the first level is cluster selection, denoted via the random variable $\mathbf{v}$, and the second level is human expert selection, denoted via the random variable $\mathbf{w}$.

Due to the presence of latent variables: *(i)* the index of the cluster being selected $\mathbf{v}$, and *(ii)* the index of the expert (either human or classifier) being selected $\mathbf{w}$, evaluating the likelihood in Eq. (34) is difficult. We, therefore, employ the EM algorithm to optimise. In the EM algorithm, we first define completed-data log-likelihood as follows:

$$Q_n = \mathbb{E}_{\Pr\left( \mathbf{v}_n, \mathbf{w}_n | \mathbf{x}_n, \mathbf{y}_n, \prod_{m=1}^{M+1} \hat{\mathbf{y}}_n^{(m)}, \gamma^{(\tau)}, \Omega \right)} \left[ \ln \Pr\left( \mathbf{y}_n, \prod_{m=1}^{M} \hat{\mathbf{y}}_n^{(m)}, \mathbf{v}_n, \mathbf{w}_n \middle| \mathbf{x}_n, \gamma, \Omega \right) \right]. \quad (35)$$

**E step**  calculates the posterior of latent variables:

$$\Pr\left( \mathbf{v}_n, \mathbf{w}_n \middle| \mathbf{x}_n, \mathbf{y}_n, \prod_{m=1}^{M+1} \hat{\mathbf{y}}_n^{(m)}, \gamma^{(\tau)}, \Omega \right) \propto \Pr\left( \mathbf{y}_n \middle| \prod_{m=1}^{M+1} \hat{\mathbf{y}}_n^{(m)}, \mathbf{w}_n \right) \Pr(\mathbf{w}_n | \mathbf{v}_n, \Omega) \Pr\left( \mathbf{v}_n | \mathbf{x}_n, \gamma^{(\tau)} \right).$$

**M step**  maximises the completed-data log-likelihood w.r.t. $\gamma$ as: $\gamma^{(\tau+1)} \leftarrow \frac{1}{N'} \sum_{i=1}^{N'} Q_n$.

Compare to Probabilistic-L2D (Nguyen et al., 2025), our formulation is simpler because the posteriors of both $\mathbf{v}$ and $\mathbf{w}$ can be obtained exactly in closed-form.

## H  Constrain workload assignment (or posterior of latent variables) to human experts

As mentioned in Section 5.2, instead of constraining on the clusters of experts, we constraint the workload assignment over all experts in all the clusters. The purpose is to have a fair comparison with the baselines.

The constrained optimisation to control workload across all experts (including both human and the classifier) can be written as follows:

$$\{q^*(\mathbf{v}_n, \mathbf{w}_n)\}_{n=1}^{N'} = \underset{q}{\arg\min} \frac{1}{N'} \sum_{n=1}^{N'} \mathrm{KL}\left[ q(\mathbf{v}_n, \mathbf{w}_n) \left\| \Pr\left( \mathbf{v}_n, \mathbf{w}_n \middle| \mathbf{x}_n, \mathbf{y}_n, \prod_{m=1}^{M+1} \mathbf{t}_n^{(m)}, \gamma^{(\tau)}, \Omega \right) \right. \right]$$

$$\text{s.t.: } \boldsymbol{\varepsilon}_l \preceq \frac{1}{N'} \sum_{n=1}^{N'} \sum_{\mathbf{v}_n=1}^{K} q(\mathbf{v}_n, \mathbf{w}_n) \preceq \boldsymbol{\varepsilon}_u.$$

$$(36)$$

We apply the Lagrangian to solve the constrained optimisation in Eq. (36). The Lagrangian of the constrained optimisation can be written as follows:

$$\mathsf{L} = \frac{1}{N'} \sum_{n=1}^{N'} \mathrm{KL}\left[ q(\mathbf{v}_n, \mathbf{w}_n) \left\| \Pr\left( \mathbf{v}_n, \mathbf{w}_n \middle| \mathbf{x}_n, \mathbf{y}_n, \prod_{m=1}^{M+1} \mathbf{t}_n^{(m)}, \gamma^{(\tau)}, \Omega \right) \right. \right]$$

$$+ \boldsymbol{\lambda}_u^\top \left[ \frac{1}{N'} \sum_{n=1}^{N'} \sum_{\mathbf{v}_n=1}^{K} q(\mathbf{v}_n, \mathbf{w}_n) - \boldsymbol{\varepsilon}_u \right] + \boldsymbol{\lambda}_l^\top \left[ \boldsymbol{\varepsilon}_l - \frac{1}{N'} \sum_{n=1}^{N'} \sum_{\mathbf{v}_n=1}^{K} q(\mathbf{v}_n, \mathbf{w}_n) \right]$$

$$= \frac{1}{N'} \sum_{n=1}^{N'} \sum_{\mathbf{z}_n=1}^{K} \sum_{\mathbf{w}_n=1}^{M+1} q(\mathbf{v}_n, \mathbf{w}_n) \left[ \ln q(\mathbf{v}_n, \mathbf{w}_n) - \ln \Pr\left( \mathbf{v}_n, \mathbf{w}_n \middle| \mathbf{x}_n, \mathbf{y}_n, \prod_{m=1}^{M+1} \mathbf{t}_n^{(m)}, \gamma^{(\tau)}, \Omega \right) \right]$$

$$+ \boldsymbol{\lambda}_u^\top \left[ \frac{1}{N'} \sum_{n=1}^{N'} \sum_{\mathbf{v}_n=1}^{K} q(\mathbf{v}_n, \mathbf{w}_n) - \boldsymbol{\varepsilon}_u \right] + \boldsymbol{\lambda}_l^\top \left[ \boldsymbol{\varepsilon}_l - \frac{1}{N'} \sum_{n=1}^{N'} \sum_{\mathbf{v}_n=1}^{K} q(\mathbf{v}_n, \mathbf{w}_n) \right],$$

$$(37)$$

where: $\boldsymbol{\lambda}_u$ and $\boldsymbol{\lambda}_l \in \mathbb{R}_+^{M+1}$ are the Lagrange multipliers.

Taking the derivative w.r.t. each element in the matrix $q(\mathbf{v}_n = k, \mathbf{w}_n = m)$ with $k \in \{1, \dots, K\}$ and $m \in \{1, \dots, M_1\}$ gives:

$$\frac{\partial \mathsf{L}}{\partial q(\mathbf{v}_n = k, \mathbf{w}_n = m)}$$

$$= \frac{1}{N'} \left[ \ln q(\mathbf{v}_n = k, \mathbf{w}_n = m) - \ln \Pr\left( \mathbf{v}_n = k, \mathbf{w}_n = m \middle| \mathbf{x}_n, \mathbf{y}_n, \prod_{m=1}^{M+1} \mathbf{t}_n^{(m)}, \gamma^{(\tau)}, \Omega \right) \right.$$

$$+ 1 + \boldsymbol{\lambda}_{um} - \boldsymbol{\lambda}_{lm} \Big].$$

$$(38)$$

Applying the KKT condition gives:

$$\frac{\partial \mathsf{L}}{\partial q(\mathbf{v}_n = k, \mathbf{w}_n = m)} = 0$$

$$\Leftrightarrow \ln q(\mathbf{v}_n = k, \mathbf{w}_n = m) - \ln \Pr\left( \mathbf{v}_n = k, \mathbf{w}_n = m \middle| \mathbf{x}_n, \mathbf{y}_n, \prod_{m=1}^{M+1} \mathbf{t}_n^{(m)}, \gamma^{(\tau)}, \Omega \right)$$

$$+ 1 + \boldsymbol{\lambda}_{um} - \boldsymbol{\lambda}_{lm} = 0$$

$$\Leftrightarrow \ln q(\mathbf{v}_n = k, \mathbf{w}_n = m) \propto \ln \Pr\left( \mathbf{v}_n = k, \mathbf{w}_n = m \middle| \mathbf{x}_n, \mathbf{y}_n, \prod_{m=1}^{M+1} \mathbf{t}_n^{(m)}, \gamma^{(\tau)}, \Omega \right)$$

$$- 1 - \boldsymbol{\lambda}_{um} + \boldsymbol{\lambda}_{lm}.$$

$$(39)$$

Or:

$$q(\mathbf{v}_n = k, \mathbf{w}_n = m) = \frac{1}{Z(\boldsymbol{\lambda}_u, \boldsymbol{\lambda}_l)} \frac{\Pr\left( \mathbf{v}_n = k, \mathbf{w}_n = m \middle| \mathbf{x}_n, \mathbf{y}_n, \prod_{m=1}^{M+1} \mathbf{t}_n^{(m)}, \gamma^{(\tau)}, \Omega \right)}{\exp(\boldsymbol{\lambda}_{um} - \boldsymbol{\lambda}_{lm} + 1)},$$

$$(40)$$

where: $Z(\boldsymbol{\lambda}_u, \boldsymbol{\lambda}_l)$ is the normalisation constant defined as follows:

$$Z(\boldsymbol{\lambda}_u, \boldsymbol{\lambda}_l) = \sum_{k=1}^{K} \sum_{m=1}^{M+1} \frac{\Pr\left( \mathbf{v}_n = k, \mathbf{w}_n = m \middle| \mathbf{x}_n, \mathbf{y}_n, \prod_{m=1}^{M+1} \mathbf{t}_n^{(m)}, \gamma^{(\tau)}, \Omega \right)}{\exp(\boldsymbol{\lambda}_{um} - \boldsymbol{\lambda}_{lm} + 1)}.$$

$$(41)$$

Substituting $q(\mathbf{v}_n = k, \mathbf{w}_n = m)$ obtained in Eq. (40) into the Lagrangian in Eq. (37) gives:

$$\mathsf{L} = \frac{1}{N'} \sum_{n=1}^{N'} \sum_{k=1}^{K} \sum_{m=1}^{M+1} q(\mathbf{v}_n = k, \mathbf{w}_n = m) \left[ -\ln Z(\boldsymbol{\lambda}_u, \boldsymbol{\lambda}_l) - \boldsymbol{\lambda}_{um} + \boldsymbol{\lambda}_{lm} - 1 \right]$$

$$+ \frac{1}{N'} \sum_{n=1}^{N'} \sum_{k=1}^{K} \sum_{m=1}^{M+1} (\boldsymbol{\lambda}_{um} - \boldsymbol{\lambda}_{lm}) q(\mathbf{v}_n = k, \mathbf{w}_n = m) - \boldsymbol{\lambda}_{um} \boldsymbol{\varepsilon}_{um} + \boldsymbol{\lambda}_{lm} \boldsymbol{\varepsilon}_{lm}. \quad (42)$$

Or:

$$\mathsf{L} = -\boldsymbol{\lambda}_u^\top \boldsymbol{\varepsilon}_u + \boldsymbol{\lambda}_l^\top \boldsymbol{\varepsilon}_l - 1 - \frac{1}{N'} \sum_{n=1}^{N'} \ln Z(\boldsymbol{\lambda}_u, \boldsymbol{\lambda}_l)$$

$$= -\boldsymbol{\lambda}_u^\top \boldsymbol{\varepsilon}_u + \boldsymbol{\lambda}_l^\top \boldsymbol{\varepsilon}_l - 1 - \frac{1}{N'} \sum_{n=1}^{N'} \ln \sum_{k=1}^{K} \sum_{m=1}^{M+1} \frac{\Pr\left(\mathbf{v}_n = k, \mathbf{w}_n = m \,\Big|\, \mathbf{x}_n, \mathbf{y}_n, \prod_{m=1}^{M+1} \mathbf{t}_n^{(m)}, \gamma^{(\tau)}, \Omega\right)}{\exp(\boldsymbol{\lambda}_{um} - \boldsymbol{\lambda}_{lm} + 1)}. \quad (43)$$

According to the duality, the Lagrange multipliers $\boldsymbol{\lambda}_u$ and $\boldsymbol{\lambda}_l$ can be obtained by maximising the Lagrangian, which is equivalent to the following:

$$\boldsymbol{\lambda}_u^*, \boldsymbol{\lambda}_l^* = \operatorname*{argmin}_{\boldsymbol{\lambda}_u, \boldsymbol{\lambda}_l \geq 0} \boldsymbol{\lambda}_u^\top \boldsymbol{\varepsilon}_u - \boldsymbol{\lambda}_l^\top \boldsymbol{\varepsilon}_l + 1$$

$$+ \frac{1}{N'} \sum_{n=1}^{N'} \ln \sum_{k=1}^{K} \sum_{m=1}^{M+1} \frac{\Pr\left(\mathbf{v}_n = k, \mathbf{w}_n = m \,\Big|\, \mathbf{x}_n, \mathbf{y}_n, \prod_{m=1}^{M+1} \mathbf{t}_n^{(m)}, \gamma^{(\tau)}, \Omega\right)}{\exp(\boldsymbol{\lambda}_{um} - \boldsymbol{\lambda}_{lm} + 1)}. \quad (44)$$

The optimisation can then be obtained by applying an off-the-shelf optimisation solver, such as non-negative projection.

# I   ALGORITHM TO CLUSTER HUMANS ON ANONYMOUS ANNOTATION DATA

---

**Algorithm 1** Learning to defer with anonymous annotation data

---

1: **procedure** MODEL-ANONYMOUS-ANNOTATION-DATA($\mathcal{S} = \{(\mathbf{x}_n, \mathbf{t}_n)\}_{n=1}^N, \alpha, K$)
2:     $\triangleright$ *$\mathbf{x}_n$: input sample*                                        $\triangleleft$
3:     $\triangleright$ *$\mathbf{t}_n$: multinomial vector of anonymous annotations*     $\triangleleft$
4:     $\triangleright$ *$\alpha$: the concentration parameter of the Dirichlet prior*     $\triangleleft$
5:     $\triangleright$ *$K$: number of clusters*     $\triangleleft$
6:     initialise $K$ parameters $(\theta_k^{(0)})_{k=1}^K$
7:     **for** $\tau \in \{0, \dots, T-1\}$ **do**           $\triangleright$ *a total of $T$ iterations*
8:        **for** $n \in \{1, \dots, N\}$ **do**
9:          E-step: $(\rho_n^*, (\zeta_{nm}^*)_{m=1}^{M_n}) \leftarrow \min$ VARIATIONAL FREE ENERGY     $\triangleright$ *Eq. (7)*
10:        calculate "complete"-data log-likelihood: $Q\left(\mathbf{x}_n, \mathbf{t}_n, (\theta_k^{(\tau)})_{k=1}^K, (\theta_k)_{k=1}^K\right)$     $\triangleright$ *Eq. (5)*
11:        M-step: $(\theta_k^{(\tau+1)})_{k=1}^K \leftarrow \operatorname{argmax}_{(\theta_k)_{k=1}^K} 1/N \sum_{n=1}^N \mathbb{E}_q\left[\ln \Pr(\mathbf{t}_n | \mathbf{x}_n, \mathbf{z}_n, (\theta_k)_{k=1}^K)\right]$.     $\triangleright$ *Eq. (8)*
12:     **return** $(\theta_k^{(T)})_{k=1}^K$

13: **procedure** INFER-CLUSTER-ASSIGNMENT-PROBABILITY($\mathcal{S}', \alpha, (\theta_k)_{k=1}^K$)
14:     $\triangleright$ *$\mathcal{S}'$: a small set of data annotated by a human expert*     $\triangleleft$
15:     $\triangleright$ *$\alpha$: an initial value of Dirichlet concentration parameter*     $\triangleleft$
16:     $\triangleright$ *$\theta_k$: parameter of each "cluster" component*     $\triangleleft$
17:     $\alpha' \leftarrow \operatorname{argmax}_\alpha \sum_{n=1}^{N'} \ln \Pr(\mathbf{t}_n' | \mathbf{x}_n', \alpha, (\theta^{(T)})_{k=1}^K)$     $\triangleright$ *Eq. (2)*
18:     calculate cluster-assigned probability: $\overline{\alpha} \leftarrow \alpha'/\mathbf{1}^\top \alpha'$     $\triangleright$ *Eq. (3)*
19:     **return** $\omega$

---

## J   DATASETS AND HYPER-PARAMETERS

### J.1   DATASETS

**Cifar-100**   consists of 50,000 training and 10,000 testing images which are grouped into 20 super-classes. Each of these superclasses consists of 5 sub-classes, making up a total of 100 classes for the whole dataset. We follow the approach of (Hemmer et al., 2023; Nguyen et al., 2025) to syn-thetically generate three pairs of experts, each exhibiting instance-dependent label noise (Xia et al., 2020). In particular, each synthetic expert in one pair would mis-classify with an error of 5, 20 and 35 percent. The reason for these error rates is to ensure that synthetic human performance is still higher than a classifier (the accuracy of the classifier is approximately 67 percent). And because the original Cifar-100 testing set has about 10 percent of duplications or almost-duplications to the ones in the training set, we employ ciFAIR-100 (Barz & Denzler, 2020), which replaces those duplicated images by ones that belong to the same classes sampled from the ImageNet dataset.

**dopanim**   is a small subset of the notorious dataset iNaturalist. The dataset consists of 15,750 ani-mal images of 15 classes, grouped into four groups of doppelganger animals and collected together with ground truth labels (Herde et al., 2024). The dataset is split into a training subset consisting of 10,500 images, and a validation subset consisting of the remaining images. We follow a similar strategy in Cifar-100 to simulate three pair of human annotators, each pair classifies with a mistake of 5, 20 and 35 percent, respectively.

**Chaoyang**   is pathology image dataset of colon patches obtained from the Chaoyang hospital in China (Zhu et al., 2022). It consists of 4,021 samples for training and 2,139 samples for testing. Three professional pathologists provide annotations over four categories: normal, serrated, adeno-carcinoma, and adenoma with the ground truth labels obtained via the majority vote of the three annotations. The performance of the three pathologists are 91, 87 and 99 percent accurate when considering majority vote as the ground truth.

### J.2   EXPERIMENT SETTING

In general, we follow the same experiment settings in previous L2D studies (Nguyen et al., 2025) to evaluate our proposed methods.

**Anonymous and identified subsets**   For each dataset, we randomly split the training set into train-ing and validation. In particular, in Cifar-100, we use 40,000 samples for training, while preserving 10,000 samples as either the context set for L2D-Pop and EA-L2D or identified set for our method. In Chaoyang, we randomly sample 500 samples without replacement to be the context/identified set. The remaining samples are used as anonymously-annotated data to train the LDA-based model. During the evaluation, we randomly select $N'$ samples out of those hold-out samples as annotator-identified data to (i) infer the cluster probability for each expert, and (ii) perform L2D-Clusters.

**Models**   Preact-Resnet-18 is used on the dataset Cifar-100, while the conventional Resnet-18 is used on the datasets dopanim and Chaoyang.

**Hyper-parameters**   For batch size, we use 128 samples on Cifar-100, and 32 samples on Chaoyang and Dopanim and optimise via stochastic gradient descend with a momentum of 0.9 and a learning of 0.001 decayed through a cosine annealing for 150 epochs, resulting in approximately 1.2 GPU-hour when running on one Nvidia GPU A6000. For the LDA-based clustering model, due to the large overlapping between the clusters of human experts (most humans annotate similarly except few cases), the Dirichlet concentration prior $\alpha$ must be large to avoid collapsing, in which any expert will be assign to the same cluster. In the experiment, we set $\alpha = 50$ for the all the datasets.

## K   ADDITIONAL CLUSTERING RESULTS

We provide additional clustering results on Cifar-100 with $K \in \{4, 5\}$ clusters in Fig. 6. In general, L2D-Pop and EA-L2D assign every synthetic human experts into the available clusters with high

confidence, although the assignment may not agree with the simulation. In contrast, our results in Figs. 6c and 6f consistently assign synthetic experts 5A and 5B into one group, and 35E and 35F into another group, while having high uncertainty assigning the remaining pair of synthetic experts 20C and 20D. Furthermore, the proposed LDA-based clustering method does not enforce to use up all the available clusters, but only a few clusters needed.

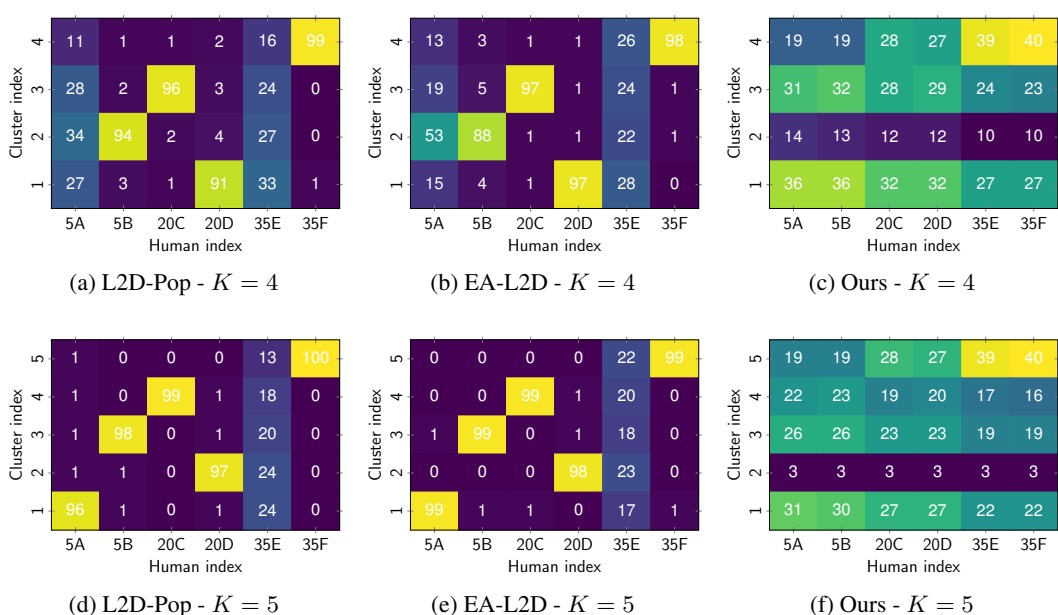

Figure 6: Additional clustering results on Cifar-100 with the context/identified set size of $N' = 100$ samples, in which the numbers in each column represent the probability of cluster assignment for each synthetic expert. Note that the colour in each heatmap is relatively scaled to enhance the visualisation.

## L    DETERMINE THE NUMBER OF CLUSTERS

We follow a similar data-driven approach in topic modelling to determine the optimal number of clusters. In particular, we calculate the *perplexity* of the hold-out anonymous annotated data, and select the setting that has the smallest perplexity score, indicating better generalisation performance.

The perplexity in our case is modified accordingly, in which it is the likelihood of the hold-out data:

$$\text{perplexity}(\mathcal{S}_{\text{val}}) = \exp\left(-\frac{\sum_{n=1}^{|\mathcal{S}_{\text{val}}|} \ln \Pr\left(\mathbf{t}_n | \mathbf{x}_n, \alpha, (\theta_k^*)_{k=1}^K\right)}{\sum_{n=1}^{|\mathcal{S}_{\text{val}}|} \mathbf{1}^\top \mathbf{t}_n}\right). \tag{45}$$

The smaller the perplexity, the better fit our model is to the data.

We train several LDA-based models with different number of clusters $K$ on the dataset Cifar-100 and plot the perplexity evaluated on a hold-out validation set in Fig. 7.

To select the optimal number of clusters $K$, we employ the heuristic metric *rate of perplexity change* (RPC) in (Zhao et al., 2015) defined as follows:

$$\text{RPC}_i = \left|\frac{P_i - P_{i-1}}{L_i - L_{i-1}}\right|, \tag{46}$$

where: $P_i$ is the perplexity evaluated with $L_i$ clusters.

The optimal number of clusters is selected as the smallest number of clusters that give the highest relative improvement:

$$K^* = \min_i \operatorname*{argmax}_i \text{RPC}_i. \tag{47}$$

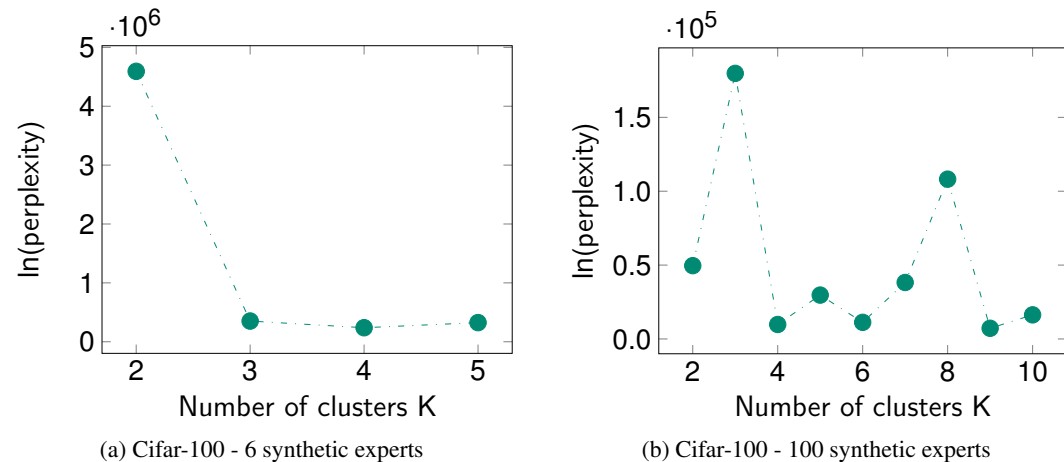

(a) Cifar-100 - 6 synthetic experts

(b) Cifar-100 - 100 synthetic experts

Figure 7: Perplexity (smaller is better) on the hold-out validation set of the dataset Cifar-100.

Hence, the optimal number of clusters for the setting of Cifar-100 with 6 and 100 synthetic experts are $K = 3$, and $K = 4$, respectively.

## M  ADDITIONAL L2D RESULTS

We provide additional results on L2D with a larger number of identified/context samples and show in Fig. 8. In this setting, the conventional human-specific L2D methods (i.e., one-stage and two-stage) have sufficient number of identified/context data to defer the decision to the most suitable experts (either human or classifier), and hence, upper-bound the performance of the remaining baselines and our method.

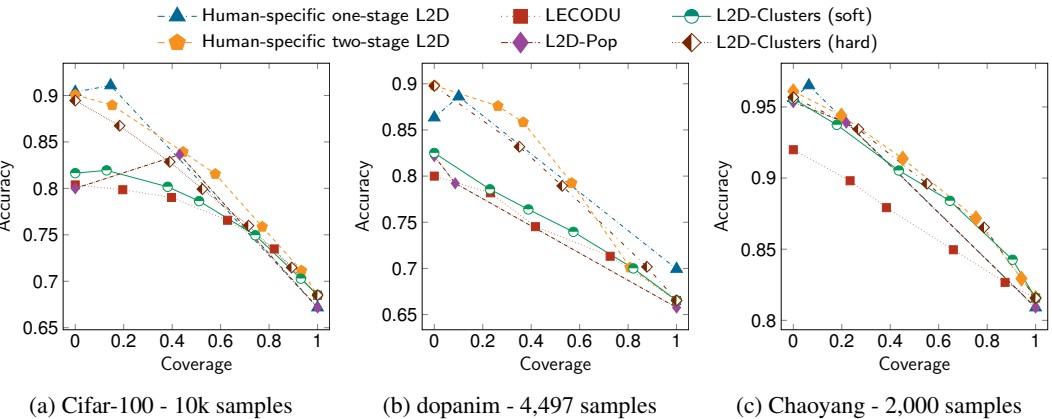

(a) Cifar-100 - 10k samples

(b) dopanim - 4,497 samples

(c) Chaoyang - 2,000 samples

Figure 8: Additional results of coverage - accuracy curves on three datasets at a large number of identified/context samples.

## N  EXPERIMENT ON LARGE NUMBER OF EXPERTS

To evaluate our proposed framework including both the LDA-based clustering and L2D-Clusters on a large number of experts, we simulate a total of 100 synthetic experts on Cifar-100. In particular, a total of 10 sets of synthetic experts is simulated, where each set consists of 10 synthetic experts having the same error rate in the set $\{0.05, 0.1, 0.15, \ldots, 0.5\}$. After performing the clustering on different number of clusters shown in Fig. 7b, $K = 4$ clusters is selected to train an L2D-Clusters system. We follow a similar setting presented in Section 5.2 for the evaluation.

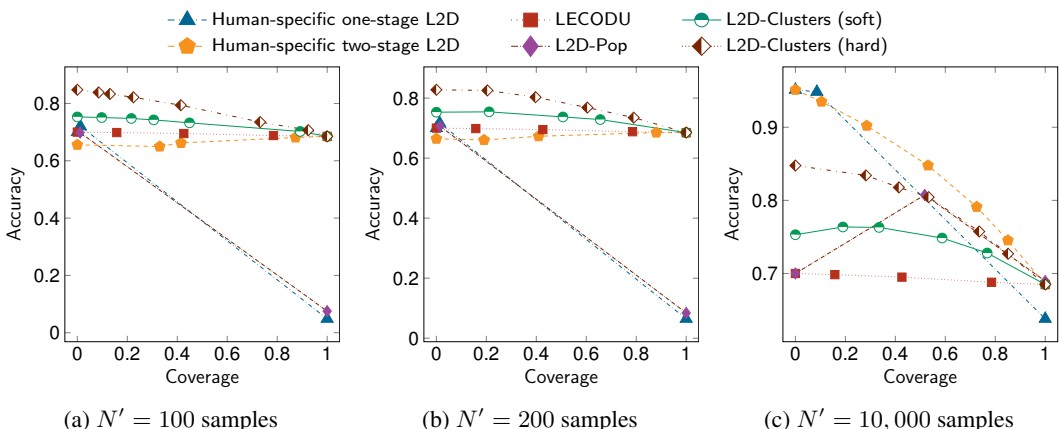



(a) $N' = 100$ samples     (b) $N' = 200$ samples     (c) $N' = 10,000$ samples



Figure 9: The accuracy - coverage curves of different methods evaluated on different sizes of the context sets in the setting consisting of 100 synthetic experts simulated in Cifar-100.

The results in Fig. 9 generally agree with our previous results evaluated on 6 synthetic experts. Specifically, our proposed L2D-Clusters method outperforms the baselines and state-of-the-art methods in the onboarding setting, where the context set size $N'$ is small, such as $N' = 100$ as in Fig. 9a and $N' = 200$ as in Fig. 9b. In these cases, both the *soft* and *hard* variants of L2D-Clusters outperform the baselines and existing methods by large margins. The performance of both one-stage L2D and L2D-Pop is significantly degraded, in which the prediction accuracy of the classifier integrated into their model is below 10 percent. This is mainly due to their single-stage nature making the integrated classifier and the gating model overfit to the small number of context set. The two-stage L2D baseline performs better than the two-stage-like methods due to its modularity using a classifier trained on ground truth data, but its performance is even worse than the random selection LECUDO method. However, when the context set size is large, such as $N' = 10,000$ as in Fig. 9c, there is sufficient amount of data to identify the characteristics of each expert, and hence, the single-stage and two-stage L2D methods perform better than L2D-Clusters, at least better than L2D-Clusters soft, due to their human-specific nature.

