# OpenReview forum: "Learning to Defer on Anonymously Annotated Data"
_ICLR.cc/2026/Conference — Submitted to ICLR 2026_

### Official Review · Reviewer_UcGT · 2025-10-15

**Soundness:** 3
**Presentation:** 3
**Contribution:** 3
**Rating:** 6
**Confidence:** 2

**Summary:**

This paper proposes L2D-Clusters as a framework extending L2D to anonymously annotated data.  The algorithm does not need fixed expert identities; instead, it uses an LDA-based model to cluster annotators by behavioral similarity and defers decisions to expert groups.

**Strengths:**

1. The paper seems to solve the limitations of existing L2D methods by introducing clustering on an anonymously labeled dataset, thereby not needing annotator identities.

2. The LDA modeling seems novel for me (though I am not familiar with L2D works).

3. The empirical results look good.

**Weaknesses:**

I am not an expert in L2D, but I feel that the model seems to be a little impractical. The model assumes that all experts in one cluster share the same labeling pattern $h(x, \theta_z)$, which is a probability vector allocating fixed probabilities to different labels to $x$. Is this too simplified? This assumption seems strong, as real annotators can demonstrate individual variability or context-dependent noise.

**Questions:**

How sensitive is your method to heterogeneity within one cluster?

---

> ### Author Response · Authors · 2025-11-20
> **Sensitivity to heterogeneity within one cluster**
>
> We appreciate the reviewer's concern and would like to clarify that our model does not assume that all experts within a cluster behave identically. On the contrary, our approach is explicitly designed to accommodate individual variability and context-dependent noise through its probabilistic clustering mechanism. In particular, each expert is assigned a soft distribution over clusters (see Eqs. (3) and (4) and Figure 2), rather than a hard cluster membership. This means that an expert's behaviour is modelled as a mixture over multiple cluster-level annotation patterns, allowing the model to capture nuanced and overlapping annotation tendencies. This design choice directly addresses the concern about oversimplification and enables the model to represent heterogeneity in expert behaviour more flexibly than prior methods.
>
> To evaluate the variability and robustness influenced by the heterogeneity of human experts within each cluster, we adopt the strategy *L2D-Clusters (soft)*, where the accuracy is the weighted average over all experts in the selected cluster as follows (the formula is already specified at line 360 in the submission):
> \\[
> \operatorname{accuracy-soft} (\text{cluster} = k) = \sum_{m = 1}^{M + 1} \Omega_{km} \Bbb{1}(\mathbf{t}^{(m)} = \mathbf{y}),
> \\]
> where $\Omega_{km}$ is the probability that the expert $m$ is in cluster $k$.
>
> The performance of this strategy in Table 2 demonstrates that our method is robust against the heterogeneity of human experts within each cluster.

---

> > ### Comment · Reviewer_UcGT · 2025-11-21
> >
> > Thanks for your response. i will keep my score.

---

### Official Review · Reviewer_J7mm · 2025-10-28

**Soundness:** 2
**Presentation:** 2
**Contribution:** 2
**Rating:** 4
**Confidence:** 3

**Summary:**

This paper tackles the "out-of-distribution problem" for human experts, where the available annotators change between training and deployment. The proposed method addresses this by learning to defer tasks to clusters of behaviorally similar experts, identified from anonymously-annotated data, rather than to specific individuals.

**Strengths:**

1. The problem of "out-of-distribution" for human experts is interesting
2. The proposed method is simple, intuitive, and effective

**Weaknesses:**

1. The paper does not provide a theoretical analysis to substantiate the method's effectiveness, such as performance guarantees or robustness analysis.
2. The performance is critically dependent on the quality of the expert clustering, a dependency amplified by the stochastic nature of randomly selecting an expert for the final decision. The validation of this clustering is confined to a limited set of datasets, and its efficacy in diverse, real-world scenarios remains unverified.
3. The empirical evaluation relies heavily on synthetic experts. The conclusions would be significantly more compelling with the inclusion of a real human study.

**Questions:**

See weaknesses

---

> ### Author Response · Authors · 2025-11-20
> **Theoretical analysis on performance guarantees or robustness**
>
> Our two-stage method are guaranteed to converge in both stages:
>  - *LDA-based clustering:* As noted in Remark 1 (Section 3.1), the LDA model is equivalent to a stochastic natural gradient algorithm (Sato, 2001) and is guaranteed to converge to a stationary point. In addition, its parameter inference relies on the EM algorithm, which ensures monotonic improvement of the log-likelihood.
>  - *L2D-Clusters* Similarly, the EM-based parameter inference guarantees that the log-likelihood does not decrease across iterations.
>
> While these properties ensure stable training, we agree that formal robustness or performance guarantees are not included in this work. We did acknowledge this as a limitation in the conclusion and plan to address it in future work.

---

> ### Author Response · Authors · 2025-11-20
> **Performance depends on clustering results and remains unverified on real-world datasets**
>
> Clustering quality is indeed a critical factor for the performance of L2D-Clusters, and we address this directly through a comprehensive evaluation in Section 5.1. Importantly, our LDA-based clustering approach consistently outperforms state-of-the-art methods such as L2D-Pop and EA-L2D, both qualitatively and quantitatively, as demonstrated in Figures 2 and 6 and Table 1. These results hold across a range of settings, including challenging synthetic scenarios with heterogeneous annotators and instance-dependent noise. Our method achieves superior clustering quality as measured by standard metrics like *adjusted Rand index* (ARI) and *adjusted normalised mutual information* (ANMI), which directly contributes to improved downstream performance.
>
> Moreover, L2D-Clusters is designed to be robust to imperfect clustering through its use of soft cluster assignments. Rather than relying on hard cluster labels, our method leverages the full posterior over cluster memberships, enabling it to distribute deferral decisions across multiple experts within a cluster.  This probabilistic routing mitigates the impact of occasional misclusterings and enhances stability.
>
> We validate the robustness of this probabilistic routing in our experiments on real-world datasets such as Chaoyang. Despite the inherent noise and variability in the expert annotations of real-world datasets, our method maintains strong performance, underscoring its practical utility and resilience. These results confirm that while clustering quality is indeed influential, our approach not only achieves state-of-the-art clustering but also gracefully handles its imperfections.

---

> ### Author Response · Authors · 2025-11-20
> **Synthetic datasets and real human study**
>
> We would like to clarify that our empirical evaluation does include real human expert data, specifically through the Chaoyang dataset. This dataset involves annotations from real-world domain experts, and in particular medical professionals, and serves as strong evidence of our method's practical applicability and robustness in real settings.
>
> The synthetic benchmarks we use are carefully designed to reflect realistic annotation conditions, such as instance-dependent label noise and heterogeneous expert behaviour, and they allow for controlled, repeatable analysis of model behaviour under varied conditions.
>
> Together, our results on both synthetic and real-world datasets provide a comprehensive and rigorous validation of our approach. The inclusion of real expert data strengthens our conclusions and demonstrates that L2D-Clusters performs effectively not only in simulated environments but also in complex, real-world annotation scenarios.

---

### Official Review · Reviewer_fVfL · 2025-11-03

**Soundness:** 3
**Presentation:** 3
**Contribution:** 3
**Rating:** 6
**Confidence:** 3

**Summary:**

The paper tries to fix a realistic weakness in learning-to-defer systems, where a model decides whether to answer or pass a case to a human. Usual methods assume the same named experts appear during training and testing, which is unrealistic. Here, the authors learn groups of similar annotators from anonymous label counts using a simple topic-model approach. When a new person arrives, the system observes a few of their labels and assigns them to one or more groups. At prediction time, the model either answers by itself or defers to one of these groups and picks any available person within it. A simple control limits how many cases go to humans.

The novelty is that the method works without knowing annotator identities or needing ground-truth labels. It also introduces a realistic onboarding scenario with many experts, each labeling only a few examples, and with noise that depends on the input itself. Experiments on CIFAR-100, dopanim, and Chaoyang datasets show that this group-based approach beats both person-specific and population-based baselines when little data is available for each expert. Once every expert has plenty of labeled data, the standard person-specific systems become stronger. Hard clustering works best overall.

**Strengths:**

The paper tackles a relevant gap in L2D. The anonymous-data LDA modeling is appropriate for multinomial label counts and cleanly separates training of clusters from later per-expert assignment using a small identified set, which reduces the dependence on an available ground truth. The L2D-Clusters architecture is a natural hierarchical mixture of experts with a gating function over clusters and random expert selection within a cluster, to wihch a workload constraint is added.
The onboarding benchmark captures conditions with many experts and few labels per expert with instance-dependent noise; it shows compelling improvements over expert-specific and population-based baselines in the small-data regime.
Writing is clear enough to follow the modeling and training details, and appendices provide derivations and the algorithm used in practice.

**Weaknesses:**

The L2D-Clusters section assumes all experts annotate the same training samples to avoid latent-variable complications but, as far as I can see, it weakens the generality of the claim about handling missing annotations and, in my view, should be mentioned more clearly in the paper.
Randomly picking an expert inside a cluster may introduce variance across the humans are subjected to---an alternative within-cluster selection rule could be investigated.
Experientally, some ablation studies are missing to assess the isolated contribution of workload constraint, online-EM momentum, or hard vs soft assignments.

**Questions:**

How sensitive are your results to the Dirichlet concentration prior for cluster mixtures? How sensitive are they to the choice of the parameter K? Can you report the performance variability (and, possibly, clustering stability/variaiblity) on an alpha-K grid?

Can the simplifying assumption that all experts annotate the same samples be relaxed?

Could you comment on any class-imbalance issues that may arise with your method?

---

> ### Author Response · Authors · 2025-11-20
> **Randomly picking an expert inside a cluster may introduce variance**
>
> We thank the reviewer for raising this point. We agree that the random selection of experts within a cluster could introduce variance and potentially bias evaluation. To mitigate this, we do not report performance based on random selection. Instead, we adopt two principled strategies:
>  - the soft strategy: Accuracy is computed as a weighted average over all experts in the selected cluster (see the formula at line 360):
> \\[
> \operatorname{accuracy-soft} (\text{cluster} = k) = \sum\_{m = 1}^{M + 1} \Omega_{km} \Bbb{1}(\mathbf{t}^{(m)} = \mathbf{y}),
> \\]
>  - the hard strategy: Prediction is made by the most probable expert in the cluster:
> \\[
> \operatorname{accuracy-hard}(\text{cluster} = k) = \Bbb{1} (\mathbf{t}^{(m^{\prime})} = \mathbf{y}),
> \\]
>
> where: $m^{\prime} = \operatorname*{argmax}_{m} \Omega\_{km}$ denoting the most probable expert assigned to the cluster.
>
> These strategies ensure fairness and robustness by leveraging cluster assignment probabilities rather than stochastic selection. Empirical results in Table 2 confirm both approaches perform consistently. We acknowledge that exploring alternative within-cluster selection rules (e.g., performance-based or uncertainty-aware selection) is an interesting future direction and could further improve robustness.

---

> ### Author Response · Authors · 2025-11-20
> **Sensitivity analysis on the hyper-parameters of the Latent Dirichlet Allocation model**
>
> **Number of clusters (K)**
>
> To assess sensitivity, we report AUC-AC (area under coverage-accuracy curve) for different $K$ values on Cifar-100 with 100 experts and identified set size of 100 samples (Appendix N):
>
> | K | AUC-AC |
> |---|---|
> | 2 | 73.72 $\pm$ 1.01 |
> | 3 | 74.27 $\pm$ 0.99 |
> | 4 | 74.00 $\pm$ 0.99 |
> | 5 | 74.23 $\pm$ 0.99 |
> | 6 | 74.04 $\pm$ 1.00 |
> | 7 | 74.19 $\pm$ 0.99 |
> | 8 | 74.17 $\pm$ 1.00 |
> | 9 | 74.05 $\pm$ 0.97 |
> | 10 | 74.07 $\pm$ 1.00 |
>
> Increasing the number of clusters $K$ can indeed allow the model to capture finer-grained annotation patterns within the anonymously annotated dataset. However, setting $K$ excessively high introduces redundant clusters whose probabilities remain near zero across all annotation sets, effectively making them "unused". Empirically, we observe that the dataset exhibits an intrinsic cluster structure with $K_{\text{intrinsic}} = 4$. Beyond this point, additional clusters do not contribute with meaningful information, leading to a plateau in performance.
>
> Despite that, fixing $K = K_{\text{intrinsic}}$ is not always optimal, as it may risk overfitting to the training data. To address this, our paper adopts a principled heuristic based on the *rate of perplexity change* (Zhao et al., 2015) evaluated on a held-out set (details in Appendix L). This metric provides a more robust criterion for selecting $K$. Importantly, our experiments show that performance remains stable across a range of $K$, indicating that the model is relatively insensitive to this hyperparameter.
>
> ---
>
> **Dirichlet prior parameter** ($\alpha$)
>
> We provide additional analysis to quantify clustering stability by computing the average pairwise cosine dissimilarity of the cluster assignment distributions defined in Eq. (3). This metric captures how dissimilar the cluster membership probabilities are between every pair of human experts (i.e., a higher cosine dissimilarity means a more diverse clustering). In addition, we report the AUC-AC score, which reflects the overall system performance across coverage–accuracy trade-offs.
>
> | $\alpha$  | Average pair-wise cosine dissimilarity ($\uparrow$) | AUC-AC ($\times$100 $\uparrow$)
> |---|---|---|
> | 0.1 | 0.0002 | 72.90 $\pm$ 2.00 |
> | 0.9 | 0.0009 | 73.10 $\pm$ 1.77 |
> | 2.0 | 0.0377 | 74.98 $\pm$ 1.32 |
> | 10 | 0.0406 | 75.07 $\pm$ 1.06 |
> | 50 | 0.0422 | 75.04 $\pm$ 1.25 |
>
> Further increasing $\alpha$ leads to a plateau in both the pairwise cosine dissimilarity and the performance (i.e., AUC-AC). This occurs because $\alpha$ governs the prior cluster distribution $\Pr(\mathbf{u} | \alpha)$ to learn the clusters of interest, whereas the posterior of cluster assignment for the human expert indexed by $m$ (see Section 3.2), denoted as $\Pr(\mathbf{u} | \alpha, \mathcal{I}\_{m})$, depends on the observed identified set $\mathcal{I}\_{m}$. Thus, while a larger $\alpha$ promotes uniformity of cluster distribution when learning the clusters, the posterior $\Pr(\mathbf{u} | \alpha, \mathcal{I}\_{m})$ for the human expert $m$ does not need to be uniform. If an expert’s annotations strongly favour certain clusters, the posterior will remain skewed toward those clusters despite the prior bias, which results in the plateau in both the pairwise cosine dissimilarity and AUC-AC.
>
> We will include these results in the revised submission to strengthen the robustness claims.

---

> ### Author Response · Authors · 2025-11-20
> **Relaxing the assumption that that all experts annotate the same samples**
>
> The assumption of "complete-annotation" can be relaxed to the "in-complete annotation" setting (also known as missing annotations). In fact, our proposed framework can be extended to handle missing annotations using techniques similar to Probabilistic-L2D (Nguyen et al., 2025), where unobserved annotations are treated as latent variables. We chose the complete-annotation setting for this submission primarily to:
>  - simplify the analysis and clearly isolate our contribution (cluster-based deferral), and
>  - ensure fair comparison with existing baselines, most of which assume complete annotations.
>
> Relaxing this assumption does not change the core methodology of our proposed method, but only introduces additional complexity in the step of parameter inference. We will include a discussion of this extension in the revision to clarify feasibility.

---

> ### Author Response · Authors · 2025-11-20
> **Potential issues with class-imbalanced datasets**
>
> Class imbalance can affect the proposed LDA-based clustering because in that setting, under-represented clusters are harder to be estimated, potentially leading to cluster collapse or merging. To mitigate this, the Dirichlet prior parameter $\alpha$ can be increased to encourage a more uniform prior distribution over clusters, forcing the model to learn all of the cluster equally, and hence, reducing the risk of collapsing or merging. In our experiments, we set $\alpha$ to a large value (e.g., $\alpha \ge 2$), which also minimises the effect of this issue. We will include a discussion of this design choice and its effect on robustness for the class-imbalanced data in our revision.

---

> > ### Comment · Reviewer_fVfL · 2025-11-24
> >
> > Thanks for your replies. I am satisfied with them and will discuss whether to update my score or not with the other revirewers.

---

### Author Response · Authors · 2025-12-03
**Summary of changes**

We thank all reviewers for their insightful and constructive feedback. We have have incorporated these suggestions into a revision, which has been uploaded to Open Review. All modifications are highlighted in blue colour for easy reference. Specifically, we have:
 - added the sensitive analysis on the number of clusters $K$ and the Dirichlet prior's parameter $\alpha$ and showed that both the LDA-based clustering and L2D-Clusters are robust to these two parameters,
 - clarified the assumption of "complete-annotation" setting to: (i) simplify our analysis, (ii) distinguish our contribution (cluster-based deferral) from the previous study for "missing annotation" setting (Nguyen et al. 2025), and (iii) ensure fairness to existing methods, in which most of them assume "complete annotation" setting, and
 - included a discussion on class-imbalance to the clustering and the mitigation by using a larger value of $\alpha$.

---

### Meta-Review · Area_Chair_F8sc · 2025-12-24

**Summary:**

This work studies the problem of learning to defer (L2D). The considered scenario uses anonymous experts-annotated dataset, so that flexibility can be gained in real-world implementation. The proposed approach leverages topic models to infer annotation patterns and to cluster annotators, which allows using representative annotators from the target cluster in the implementation phase.

Strengths:

The reviewers acknowledged that the identified research gap is meaningful. They also appreciated the simplicity and intuition of the proposed approach.

Weaknesses:

Two reviewers pointed out that there is a lack of theoretical contribution. They also question is relying on clustering performance would make the method sensitive to dataset or scenario changes.


Overall, the proposed approach appears to be an interesting application of topic models. The research gap is meaningful in L2D. The proposed method is intuitive and simple.  However, the proposed method lacks theoretical support and deeper understanding beyond empirical validation. The robustness is still questionable as the method relies on accuracy of clustering. The rebuttal did not satisfactorily resolve these questions.

**Reviewer Concerns:**

For the “lack of theoretical contribution” question, the authors replied with algorithm convergence and objective function decrease. These seem to be irrelevant to model robustness and generalization performance, and the authors acknowledge that there is a lack of such important guarantees.

In terms of the reliance on clustering performance, the rebuttal essentially states that empirical results on a variety of datasets supports the design. The rebuttal also mentioned that LDA is soft clustering other than hard clustering, and thus would be more robust. The AC does not agree with this argument - using soft clustering does not mean the method is robust to noise or outliers. In fact, simplex based constraints on cluster indicators make the algorithm more sensitive to noise.

**Reviewer Scores:**

Reviewer fVfL (rating 6) expressed satisfaction of the rebuttal.

Reviewer J7mm (rating 4) expressed concerns on the lack of theoretical justification, the performance relying on expert clustering, the stochastic nature of selecting an expert for final decision, and the lack of human expert based experiments. The rebuttal acknowledged that only loss decrease type guarantee can be talked about, LDA is proposed for robust non-expert clustering, and human expert was in Chaoyang data. The AC is not quite convinced by the first two points (as explained in summary). The score may not be lifted.

Reviewer UcGT (rating 6) questions practicality of the model assuming all experts sharing an encoder. The question was answered, and the reviewer expressed that they will keep the score.

---

### Decision · Program_Chairs · 2026-01-26

Reject